# Deep ocean metagenomes provide insight into the metabolic architecture of bathypelagic microbial communities

Silvia G. Acinas [1,26✉], Pablo Sánchez [1,26], Guillem Salazar [1,2,26], Francisco M. Cornejo-Castillo [1,3,26], Marta Sebastián [1,4], Ramiro Logares [1], Marta Royo-Llonch [1], Lucas Paoli[2], Shinichi Sunagawa [2], Pascal Hingamp[5], Hiroyuki Ogata [6], Gipsi Lima-Mendez [7,8], Simon Roux [9,25], José M. González [10], Jesús M. Arrieta [11], Intikhab S. Alam [12], Allan Kamau [12], Chris Bowler [13,14], Jeroen Raes[15,16], Stéphane Pesant[17,18], Peer Bork [19], Susana Agustí [20], Takashi Gojobori[12], Dolors Vaqué [1], Matthew B. Sullivan [21], Carlos Pedrós-Alió[22], Ramon Massana [1], Carlos M. Duarte [23] & Josep M. Gasol [1,24]

The deep sea, the largest ocean's compartment, drives planetary-scale biogeochemical cycling. Yet, the functional exploration of its microbial communities lags far behind other environments. Here we analyze 58 metagenomes from tropical and subtropical deep oceans to generate the Malaspina Gene Database. Free-living or particle-attached lifestyles drive functional differences in bathypelagic prokaryotic communities, regardless of their biogeography. Ammonia and CO oxidation pathways are enriched in the free-living microbial communities and dissimilatory nitrate reduction to ammonium and $H_2$ oxidation pathways in the particle-attached, while the Calvin Benson-Bassham cycle is the most prevalent inorganic carbon fixation pathway in both size fractions. Reconstruction of the Malaspina Deep Metagenome-Assembled Genomes reveals unique non-cyanobacterial diazotrophic bacteria and chemolithoautotrophic prokaryotes. The widespread potential to grow both auto-trophically and heterotrophically suggests that mixotrophy is an ecologically relevant trait in the deep ocean. These results expand our understanding of the functional microbial structure and metabolic capabilities of the largest Earth aquatic ecosystem.

A list of author affiliations appears at the end of the paper.

Most of the ocean's life is isolated from sunlight, our planet's primary energy source. Besides living in permanent darkness, deep-ocean organisms have to cope and adapt to the high pressure and low temperature that characterize this ecosystem. This fascinating habitat represents one of the largest biomes on Earth, mostly occupied by bacteria and archaea that play a pivotal role in biogeochemical cycles on a planetary scale[1,2]. Microbial metabolisms in the deep ocean have been assumed to be primarily heterotrophic, relying on organic matter exported from the sunlit layer through sinking particles (zooplankton fecal pellets, phytoplankton aggregates, and other types)[3,4]. However, the high respiratory activity measured in the dark ocean is difficult to reconcile with the rates of supply of organic carbon produced in the photic layer[3–7], suggesting the existence of other sources of carbon, such as potentially non-sinking POC and in situ production by autochthonous microbial chemolithoautotrophs[8]. Additional pathways that inject particles into the ocean interior (particle injection pumps; PIPs)[9] can be mediated by physical processes such as subduction[10,11] or biological processes, such as the daily migration of zooplankton and small fish[12,13].

Chemolithoautotrophy has been considered as a possible pathway supporting the high dark ocean respiratory activity[7,8,14–16]. Experimental rate measurements and bulk biogeochemical estimates agree with findings using single-cell genomics that ubiquitous bacterial lineages have the potential for inorganic carbon fixation[17,18], suggesting that chemolithoautotrophy may be a substantial contributor to deep-sea metabolism and may play a greater role in the global ocean carbon cycle than previously thought. Whereas inorganic carbon fixation is energetically costly[19], mixotrophy, i.e., carrying out chemolithoautotrophic inorganic carbon fixation and heterotrophy[20,21], may constitute a cost-effective strategy for microorganisms to persist in the dark ocean, although its extent in the global deep ocean is unknown.

Despite the potential importance of organic carbon derived from chemolithoautotrophy, particles likely represent the main source of reduced C to the deep ocean and constitute important hotspots of microbial activity that fuel the dark ocean food web[7]. These can be fast-sinking particles, traveling through the water column in a few weeks[22–24], or buoyant or slow-sinking organic particles, which remain suspended in the deep ocean over annual time scales[7]. The delivery of fast-sinking particles to the deep ocean depends on the trophic and functional structure of the surface ocean. As a result, this flux is intermittent[25] and heterogeneously distributed on the spatial scale, which may lead to a heterogeneous distribution in the metabolic capacities of deep-ocean microbes across the global ocean. The diversity and biogeography of bathypelagic prokaryotic communities have recently been described at a global scale, showing that the free-living (FL) and particle-attached (PA) microbial communities differ greatly in taxonomic composition[26,27] and appear to be structured by different ecological drivers[26]. The lifestyle dichotomy between FL and PA prokaryotes was shown to be a phylogenetically conserved trait of deep-ocean microorganisms[28]; however, the differences in the functional capacities of these two groups of microorganisms remain largely unexplored. Despite the existence of some studies at the local/regional scale[29–32], a global understanding of the ecology and metabolic processes of deep-sea bacterial and archaeal microorganisms similar to that available for the upper/photic ocean is still missing[33–35].

The Malaspina 2010 Circumnavigation Expedition aimed to address such knowledge gaps by surveying bathypelagic microbes in the tropical and subtropical oceans[36]. Here, we analyze 58 deep-sea microbial metagenomes from that expedition and release the Malaspina Gene DataBase (M-GeneDB), a valuable deep-ocean microbial community genomic dataset. In addition, we constructed the Malaspina Deep Metagenome-Assembled Genomes (MDeep-MAGs) catalog to explore the metabolic potential of the deep-sea microbiome.

## Results and discussion

**Gene-centric taxonomic composition of the Malaspina Gene DataBase (M-GeneDB).** The 58 microbial metagenomes were sampled between 35°N and 40°S from 32 stations (St) in the North and South Pacific and Atlantic Oceans, the Indian Ocean, and the South Australian Bight (Fig. 1a) from an average water depth of 3731 m (Fig. 1b). Two different plankton size fractions were analyzed in each station representing the FL (0.2–0.8 μm) and PA (0.8–20 μm) microbial communities, the later fraction also including bathypelagic microbial eukaryotes. A total of 195 gigabases (Gb) ($6.49 × 10^8$ read pairs) of the data with an average of 3.36 Gb per sample were generated (Supplementary Data 1). The number of predicted genes was 4.03 million (M) from the 58 assembled bathypelagic metagenomes. M-GeneDB was first built by clustering nucleotide sequences at 95% sequence identity to remove redundancy and for consistency with the *Tara* Oceans gene set, yielding 1.12 M non-redundant unique sequence clusters (referred hereafter as genes). The M-GeneDB was next integrated into the recently updated Ocean Microbial-Reference Gene Catalogue of *Tara* Oceans (OM-RGC.v2;[37] Supplementary Data 1) by further clustering both databases. The novelty of the M-GeneDB is represented by a total of 647,817 Malaspina-exclusive genes that account for 58% of the total genes (Fig. 1c) that were absent from the *Tara* Oceans global survey[32], representing a unique gene repertoire complementary to the epipelagic and mesopelagic genes reported by *Tara* Oceans[34,37] (Fig. 1b, c). The nature of the novelty of the M-GeneDB lies in its 63% of "unknown" genes without functional annotation. The remaining 37% genes were linked to transporters, including two-component systems and ABC transporters (>9.7%), followed by DNA repair and recombination proteins (2.6%) and peptidases (1.8%; Supplementary Fig. 2; Supplementary Data 2). Other genes >1% were secretion systems, aminoacidic related enzymes, or quorum sensing genes (Supplementary Fig. 2). These results agree with previous findings of the prevalence of these genes in the deep ocean[29,38–40].

Although we acknowledge that this comparison represents a snapshot of the currently known databases, this number reflects the vertical functional stratification and dichotomy between photic and aphotic microbiomes, as well as the differences between the mesopelagic and the bathypelagic. Despite the role of sinking particles in delivering epipelagic microbes to the deep-sea[27], the presence of endemic bathypelagic microorganisms has been described in several of the Malaspina sampled stations[41]. On average 61 (± 14% SD) of the predicted genes in each sample were exclusively found in the present dataset, which highlights the unique gene content of the bathypelagic microbiome (Fig. 1c and Supplementary Data 1). Each sample contained 14 ± 9% specific genes not found in any other Malaspina samples (Fig. 1c and Supplementary Data 1). Station St62 in the Indian Ocean, sampled at 2400 m, showed the highest fraction of sample-specific genes with 43% of the total being also highly different in terms of taxonomic community composition[26] (Supplementary Fig. 3). This sample, together with other four stations located in Brazil (St32), North Atlantic American (St134), and Guatemala basins (St121), that harbored more than 30% of sample-specific genes, were all from the PA size fraction and associated with circumpolar deep water and North Atlantic Deep Water masses (Supplementary Data 1).

The taxonomic affiliation of the genes in the M-GeneDB indicated that most of them belonged to the bacteria and archaea

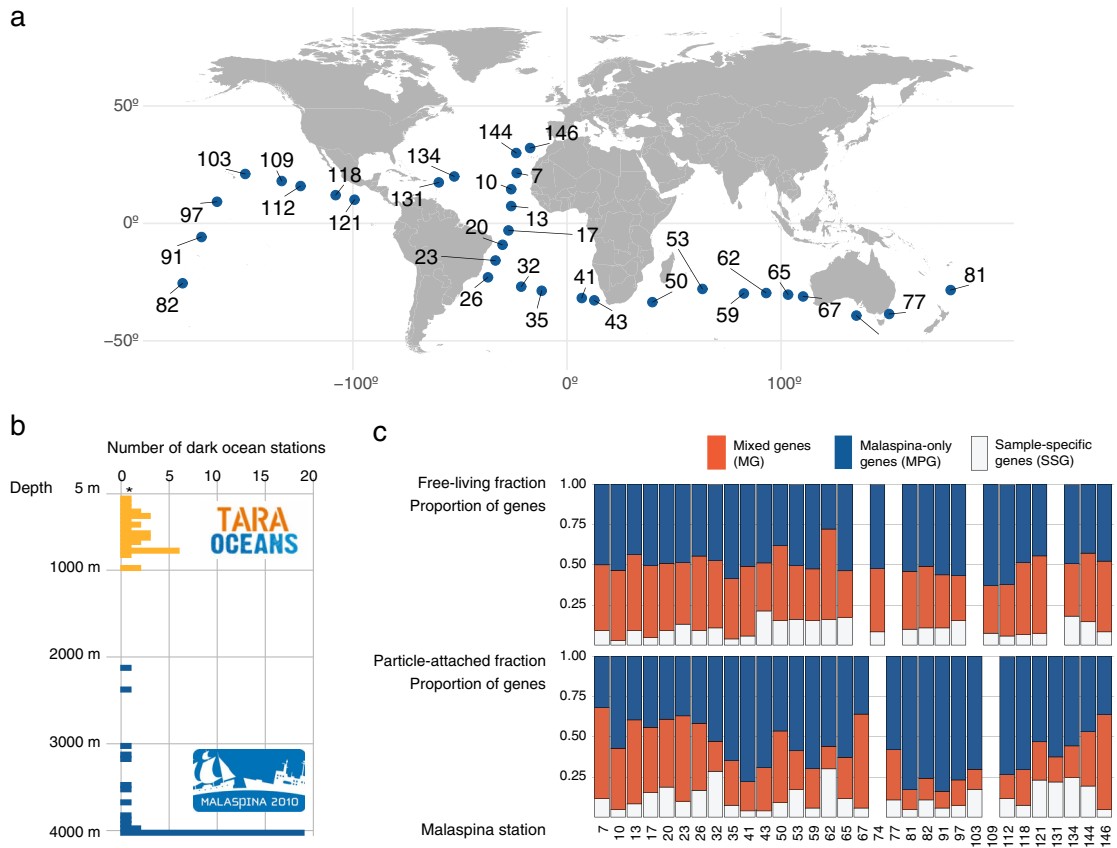

**Fig. 1 Malaspina Deep Ocean Genetic Resources. a** Malaspina 2010 expedition cruise track showing the locations of the 32 stations sampled for the present study. **b** Representation of the sampling depth and metagenomics dataset generated by the *Tara* Oceans and Malaspina 2010 Circumnavigation Expeditions. The histogram plot displays the number of stations sampled in the dark ocean during the *Tara* Oceans (orange) and Malaspina 2010 (blue) expeditions and the distribution by water depth. The Malaspina Gene Database (M-GeneDB) was generated from the integration of 58 metagenomic bathypelagic samples. The asterisk in the histogram indicates the samples collected in the photic layer in *Tara* Oceans that were not included in the figure. **c** Analyses of the integrated gene catalog that results from the *Tara* Oceans (OM-RGC.v2) and M-GeneDB. The relative abundance of unique genes that appear only in Malaspina (MPG) (solid blue), Mixed genes (MG) that are present in both catalogs (red), and the Malaspina Sample-Specific genes (SSG) in white for both the free-living (FL; 0.2–0.8 µm) and particle-attached (PA; 0.8–20 µm) fractions.

domains, with a minor representation of viruses in both size fractions and eukaryotes that were mostly found in the PA fraction (Supplementary Fig. 1). Nevertheless, a significant proportion (~25–30%) of the genes in both size fractions could not be assigned to any domain (Supplementary Fig. 1). Microbial taxonomy (i.e., small eukaryotes (Supplementary Fig. 3a), prokaryotes (Supplementary Fig. 3b), giruses (Supplementary Fig. 3c), and viruses (Supplementary Fig. 3d) in the bathypelagic samples was evaluated using different marker genes extracted from the metagenomes (Supplementary Data 3, Supplementary Data 6, see Supplementary Discussion). The identified diversity concurred with previous results based on 18S[42] and 16S rRNA[26] PCR amplicons from the same samples. Identification of nucleocytoplasmic large DNA viruses (NCLDVs) revealed their ubiquity in both size fractions and in all ocean basins (Supplementary Fig. 3c). The dominant NCLDVs in the deep ocean were *Megaviridae* (76% and 63% in the FL and PA fractions, respectively). This result contrasts with the lower proportion (36%) of *Megaviridae* in the sunlit ocean[43].

**Functional architecture of the deep-ocean microbiome.** We evaluated the effect of the different oceanic regions, basins, and lifestyles (FL or PA) on determining the bathypelagic prokaryotic functional structure (Fig. 2). Our results identified two main

functional groups of samples corresponding to PA and FL communities (Fig. 2a). This pattern was coherent regardless of the functional classifications used: the Kyoto Encyclopedia of Genes and Genomes Orthologs[44] (KOs; Fig. 2), clusters of orthologous groups[45] (COG), protein families[46] (Pfam), or Enzyme Commission numbers (EC; Supplementary Fig. 4). The PA and FL lifestyle explained 20.6 % of the variance followed by ocean basins (16.6 %) and oceans (7.4%; PERMANOVA test, $P < 0.001$). Although previous studies have also shown contrasting gene repertoires for FL and PA microbial communities, they were limited to a few studies at the local scale in the photic ocean[47,48], coastal ecosystems[49] or to the oxygen minimum zone (OMZ) off the coast of Chile[32] and a global comparison had not been presented.

These findings highlight that the main factor that functionally structures the microbial communities in the bathypelagic deep ocean is community lifestyle rather than their geographic origin, although differences among the different oceanic basins (Fig. 2b) were also observed. Thus, FL and PA prokaryotic microbial communities of the deep ocean not only represent different taxonomic groups as previously recognized[26,27] but also correspond to lifestyles with consistently different functional traits.

To explore the potential metabolic differences between FL and PA prokaryotic microbial communities (Figs. 3 and 4), a selection of marker genes (Supplementary Data 7) related to

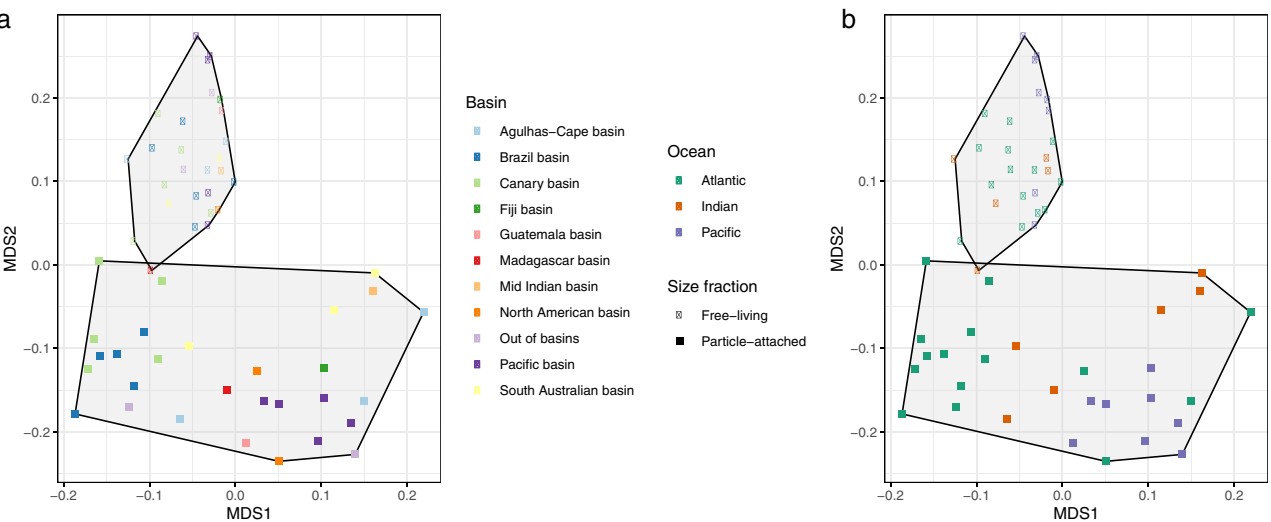

**Fig. 2 Functional community structure of the bathypelagic microbial communities.** Nonmetric multidimensional scaling (NMDS) of the microbial communities based on the functional compositional similarity (Bray–Curtis distances) among the 58 samples in the dataset, based on clusters of KEGG orthologous groups (KOs). **a** Size fraction is coded by the symbol (squares, particle-attached and circles, free-living prokaryotes) and **b** the main oceans and deep-oceanic basins by color codes (see legends).

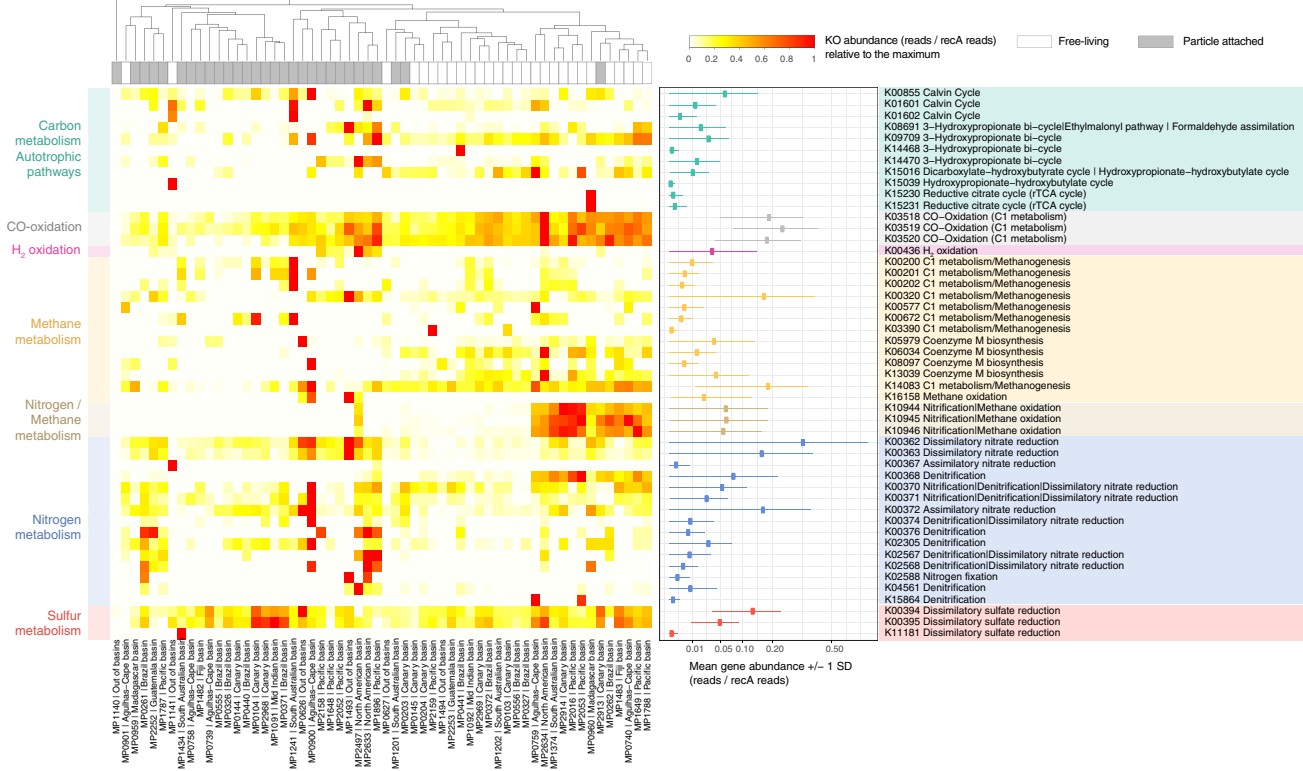

**Fig. 3 Heatmap of selected marker genes for different metabolic pathways across the 58 metagenomes.** A total of 49 marker genes (KOs, Y axis) indicative of different metabolic processes (Supplementary Data 8) detected in the Malaspina samples (X axis). KO abundance was normalized by *recA* single-copy gene as a proxy for copy number per cell. The general metabolism assignation is color-coded (see legend in the upper right) and the KEGG module(s) assignation used in the KO label is also indicated. The relative abundance across samples for each KO is shown in the heatmap. The mean (± 1 SD) untransformed abundance of each KO across all samples (reads/*recA* reads) is presented in the right panel.

potential relevant pathways in the deep ocean related to inorganic carbon fixation, nitrogen, sulfur, methane, and hydrogen metabolisms were searched in the bathypelagic metagenomic dataset (Fig. 3). Out of these key marker genes, 49 KOs were present in the M-GeneDB (Supplementary Data 8). Among them, the key genes of four different inorganic $CO_2$-fixation pathways.

The Calvin–Benson–Bassham (CBB) cycle, identified by the ribulose-bisphosphate carboxylase large subunit (RuBisCO, K01601: *rbcL*) was widely distributed in the dataset followed by the 3-HP (K14468: *mcr*, K08691: *mcl*), whereas the archaeal 3-hydroxypropionate–4-hydroxybutyrate cycle (K15039), and the rrTCA cycle (K15230: *aclA*/ K15231: *aclB*) displayed a narrow

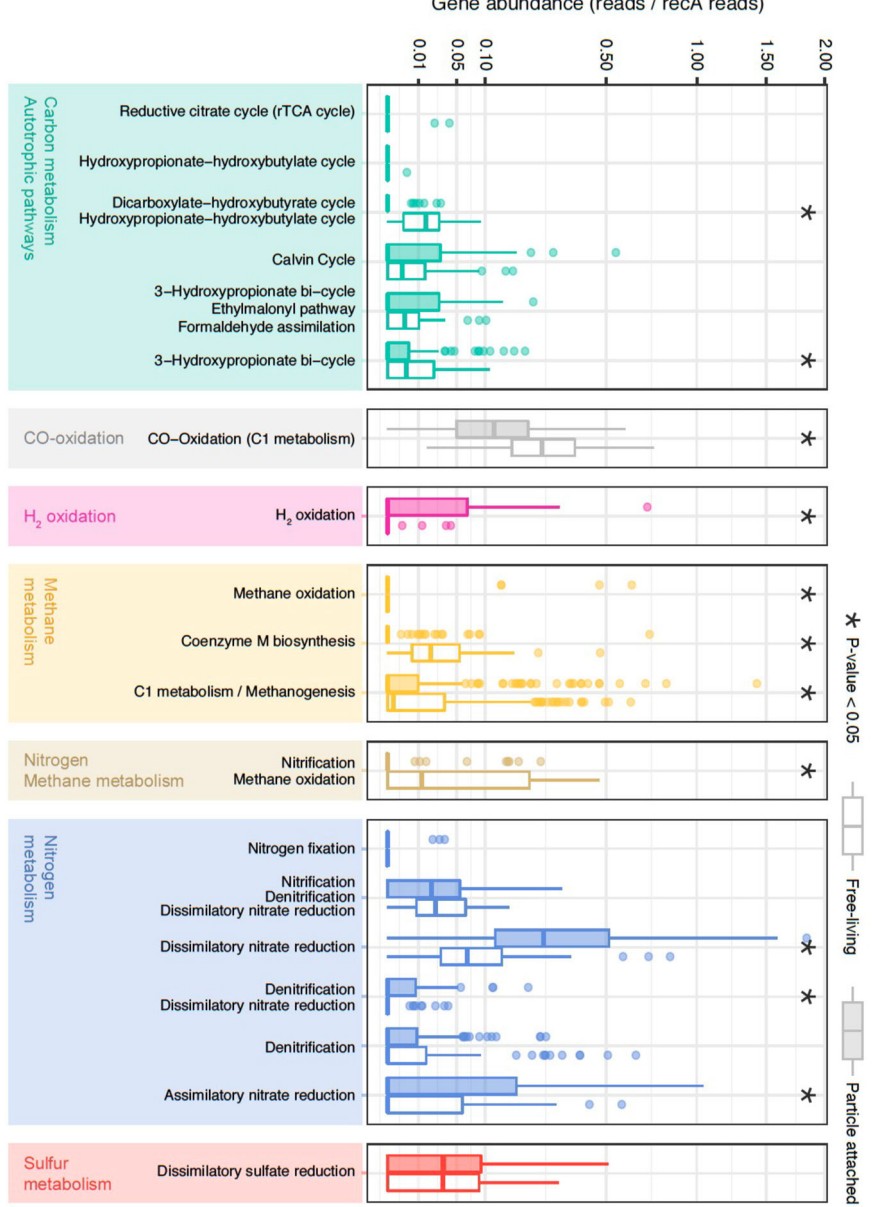

**Fig. 4 Comparison of metabolic pathways in the free-living and particle-attached microbial communities from the bathypelagic ocean.** Normalized gene abundance (reads/*recA* reads) of 49 marker genes indicative of different autotrophic carbon-fixation pathways, nitrogen, sulfur, methane, and hydrogen metabolisms in the Malaspina Gene DataBase. Gene abundances from KOs belonging to the same pathway and KEGG module level have been plotted together (Supplementary Data 7). From top to the bottom: Reductive citrate cycle (rTCA): K15230, K15231); hydroxypropionate–hydroxybutylate cycle: K15039; dicarboxylate-hydroxybutyrate cycle|hydroxypropionate-hydroxybutylate cycle: K15016; Calvin cycle: K00855; K01601 and K01602; 3-hydroxypropionate bi-cycle|ethylmalonyl pathway|formaldehyde assimilation: K08691; 3-hydroxypropionate bi-cycle: K14468, K14470 and K09709; CO oxidation (C1 metabolism): K03520; K03519 and K03518; H2-oxidation: K00436; methane oxidation: K16158; coenzyme M biosynthesis: K05979, K06034, K08097 and K13039; C1 metabolism/methanogenesis: K00320, K00200, K00201, K00202, K00672, K03390, K14083, and K00577; nitrification|methane oxidation: K10944, K10945, and K10946; Nitrogen fixation: K02588; nitrification|denitrification|dissimilatory nitrate reduction: K00370 and K00371; dissimilatory nitrate reduction (DNRA): K00362 and K00363; denitrification|dissimilatory nitrate reduction: K00374, K02567, and K02568; denitrification: K00368, K15864, K04561, K02305, and K00376; assimilatory nitrate reduction: K00367 and K00372; dissimilatory sulfate reduction: K00394, K00395 and K11181. Wilcoxon tests were done to test for significant differences between the particle-attached (PA) and free-living (FL) assemblages and significant (*P* value < 0.05) differences are labeled with asterisks. FL (empty boxes) and PA (filled boxes) bathypelagic microbial communities are shown next to each other.

distribution restricted to a single station each: St62 and St53, respectively, both in the Indian Ocean (Fig. 3). RuBisCO occurred in 48% of the samples (Supplementary Data 8), and on 1.3% of the potential cells on average (based on *recA* normalization) with two significant deviations observed in the PA fractions at St67 (South Australian Bight, Indian Ocean) and St134 (North American basin, Atlantic Ocean) where it peaked up to 12% of the cells (Fig. 3). However, there were no significant differences between both size fractions (Wilcoxon test, *P* value = 0.135).

Nitrification through ammonia[15,50,51] and nitrite oxidation[18] have been postulated as the main sources of energy for carbon fixation in the dark ocean. For ammonia oxidation, we looked for

the KEGG ortholog K10944 (*pmoA-amoA*) corresponding to the marker gene methane/ammonia monooxygenase subunit and the protein family PF12942 to identify the archaeal ammonia monooxygenase subunit A. The abundances of both markers across samples were positively correlated (Spearman correlation, $r = 0.69$, $P = 2.55\text{E-09}$), and they were found in ~36% of our samples and in 6% of microbial cells, mostly in FL microorganisms (Wilcoxon test, $P$ value < 0.005) (Supplementary Fig. 5). For nitrite oxidation, the nitrate reductase/nitrite oxidoreductase (K00370/K00371) was found in 81% of the samples and in ~4% of the microbial cells, although this enzyme could also participate in other metabolic processes as in dissimilatory nitrate reduction and denitrification (Fig. 3). Thus, exploring the co-occurrence of other key genes of each pathway in metagenomic bins is necessary for their validation as shown below. Other relevant nitrification genes, such as the hydroxylamine dehydrogenase (K10535: *hao*) or the key enzyme for the anaerobic ammonia oxidizers (K20932/K20935: *hdh*; hydrazine dehydrogenase; anammox bacteria), normally present within the oxygen minimum zones in the mesopelagic ocean, were absent from our bathypelagic metagenomic dataset. This reflects different biogeochemical processes occurring in the anoxic mesopelagic OMZ and the oxic bathypelagic oceans.

H$_2$ and CO oxidation were explored as potential alternative energy sources[52]. H$_2$-oxidation has been described in hydrothermal vents[53] or subsurface microbial communities[54] but we also found it in 24% of the samples (Fig. 3), mostly in PA microorganisms (Wilcoxon test, $P$ value <0.005; Fig. 4). These results expand the ecological niches of microbial H$_2$ oxidizers in the bathypelagic ocean, probably associated with particles providing anoxic microenvironments where H$_2$ production by fermentation is favored. The oxidation of CO is catalyzed by CO dehydrogenase (CODH; *cox* genes)[55,56] and has been associated with Actinobacterota, Proteobacteria, and members of Bacteroidota and Chloroflexota phyla[57]. The ubiquitous distribution of CO oxidation by *cox* genes (K03518: *coxS*, K03519: *cosM* and K03520: *coxL*) was notable since it was detected in 87% of the samples and with a high abundance (average 20% of the microbial cells; Fig. 3), mostly in FL prokaryotes (Wilcoxon test, $P$ value < 0.005) (Fig. 4), pointing to CO oxidation as an important energy supplement for heterotrophs in the deep ocean.

Dissimilatory nitrate reduction to ammonium (DNRA) was also detected. Due to the absence of the periplasmic pentahaem cytochrome $c$ nitrite reductase *nrfA*[58] in our dataset (Supplementary Data 7), nitrate reduction to ammonium seemed to be catalyzed by the cytoplasmic NADH-dependent nitrite reductase *nirB* or by its two-subunit variant *nirBD* (K00362: *nirB* -K00363: *nirD*)[59]. The potential DNRA and other metabolisms such as denitrification (K00368: *nirK*; K02305: *norC*; K04561: *norB*; K00376: *nosZ*) and sulfate reduction (K00394: *aprA*-K00395: *aprB*) were also present. Up to 27% of the samples contained denitrification-related genes and 90% of the samples displayed marker genes of sulfate reduction (Fig. 3 and Supplementary Data 8). The prevalence of such metabolisms in well-oxygenated waters might be explained by the formation of microenvironments inside organic aggregates or particles, where intense respiration may result in local O$_2$ exhaustion[60]. This is supported by the finding that both the assimilatory and dissimilatory nitrate reduction pathways were enriched in the PA fraction (Wilcoxon test, $P$ value < 0.005; Fig. 4). While the potential for DNRA was present in most of the samples at abundances that reached up to 34% of the potential microbial cells (Fig. 3), denitrification was less abundant (between 0.7 and 8% of microbial cells) despite being widely distributed. Finally, dissimilatory sulfate reduction genes were found in most of the FL and PA microbial communities in 5–13% of the cells (Fig. 3). Overall, we found

that the CBB cycle was distributed in both size fractions, whereas the CO oxidation or ammonia oxidation were enriched in FL and H$_2$ oxidation, and conversely DNRA, were enriched in PA microbial communities. Our results not only highlight the distribution of the main potential biogeochemical processes in the bathypelagic ocean, some of them presenting a patchy distribution, but also attribute specific metabolic pathways to either the deep-ocean FL or PA prokaryotes.

**Diversity and novelty of the Malaspina Deep MAGs catalog.** Co-assembly of the 58 bathypelagic metagenomes allowed the reconstruction of 619 non-redundant bins with 57.2% mean genome completeness, accounting for a total of 1.4 Gbp. A total of 317 of these bins had ≥50% genome completeness values and <10% contamination and fulfilled the quality standards[61] to be considered as medium or high-quality Metagenome-Assembled Genomes (MAGs). These 317 MAGs, with 84.2% average genome completeness, represented a total of 936 Mbp. This dataset is referred to here as the Malaspina Deep MAGs catalog (MDeep-MAGs). A total of 298 bacterial MAGs were taxonomically assigned to 22 phyla and 19 archaeal MAGs were assigned to 4 phyla based on the GTDB taxonomy (Fig. 5). In addition, two low-quality bins (0046 and 0224) were assigned to eukaryotes, potentially to fungal taxa. Proteobacteria ($n = 149$) followed by Bacteroidota ($n = 30$), and Chloroflexota ($n = 19$) were the most abundant bacterial phyla in the MDeep-MAGs collection, while for archaeal MAGs Thermoplasmatota ($n = 9$) and Crenarchaeota ($n = 6$) had the most representatives (Fig. 5a). The more abundant bacterial classes were Gammaproteobacteria ($n = 82$) and Alphaproteobacteria ($n = 67$). The MDeep-MAGs included a remarkable taxonomic novelty with >68% and 58% of novel species within the archaea and bacteria MAGs, respectively. Within bacteria, MAG0213 was assigned to a novel Class of the Latescibacterota phylum, one MAG may represent a novel order and five MAGs could represent distinct novel families (Fig. 5b and Supplementary Data 9). In the case of archaea, we found MAG0485 as potentially representing a novel family within the Nanoarchaeota phylum (Fig. 5b).

Interestingly, the MDeep-MAGs recruited 32% of the reads of the total free-living fraction bathypelagic metagenomes and ~20% of the reads in the PA fraction (Fig. 5c). These differences may suggest that a larger proportion of diversity is missing from the PA fraction. One possibility is due to the presence of picoeukaryotes since protists, which have larger genomes, are more fragmented in the metagenome and are more difficult to bin into MAGs, as suggested by the lower individual assembly sizes of PA compared to FL ($22.4 \pm 13.6$ and $36.9 \pm 17.1$ Mbp, respectively; Supplementary Data 1) obtained with the same sequencing depth. This lower read mapping coming from these large-size fraction samples may be due to the lower genome reconstruction of picoeukaryotes. Also, it has been shown that the prokaryotic phylogenetic diversity per OTU and the mean nearest taxon distance (MNTD) is higher for the PA fraction[28] and this may affect the lower mapping rate on the genomes present in this fraction.

Overall, these recruitments were higher than those reported for the photic layer of the global ocean *Tara* Oceans dataset (6.84% of the metagenomic reads)[62] and specifically for the Mediterranean Sea, which showed average mapping rates of 14% of the metagenomic reads for different microbial size fractions[63]. Such discrepancy may be due to probable differences in sequencing depth and methodological variations in the co-assembly, filtering, and mapping between studies.

MAG completion estimates based on domain-specific single-copy core genes[64] ranged from 50.3 to 100% (Supplementary

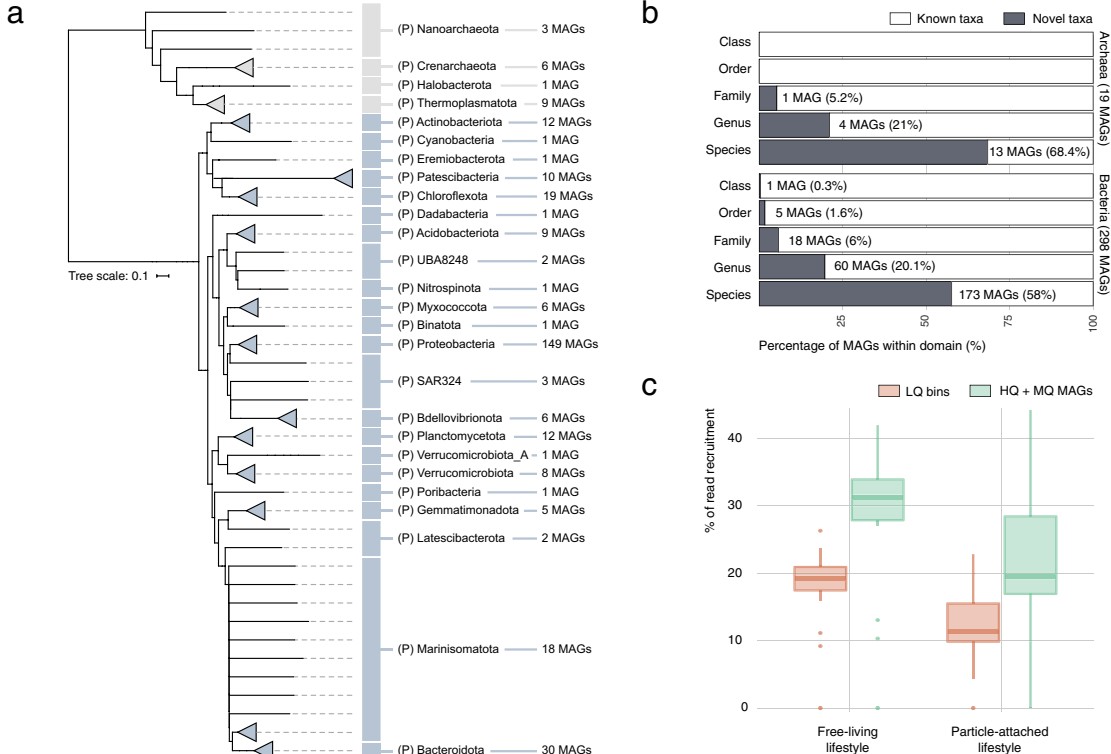

**Fig. 5 Taxonomy and novelty of the Malaspina Deep MAGs catalog (MDeep-MAGs). a** Phylogenomics-based taxonomic classification of the 317 MDeep-MAGs (i.e., high-quality bins) dataset obtained from co-assembling 58 bathypelagic ocean metagenomes. MAGs are displayed at the phylum (P = phylum) taxonomic level using the closest reference based on the Genome Taxonomy Database GTDB. **b** Stacked bar plot for novelty quantification of the Malaspina MDeep-MAGs (X axis) according to their taxonomic ranks (Y axis) for archaea and bacteria. The taxonomically unclassified portion is depicted in white and classified in gray. **c** Distribution of metagenomic reads' recovery by the low-quality metagenomic bins (LQ in orange) and the 317 MDeep-MAGs (that corresponded to medium quality (MQ) and high-quality (HQ) MAGs) reconstructed per sample in green. Samples are divided by lifestyle (free-living and particle-attached).

Data 9). The mean genome assembly size of the 317 MAGs was of 2.9 Mbp, ranging from 0.3 to 10.9 Mbp (Supplementary Data 9). The smallest genome corresponded to an archaeon (MAG0485) of the phylum Nanoarchaeota, first described as obligate symbionts with reduced genomes in marine thermal vent environments[65], but later found in a wide range of environments and temperatures. The largest genome (MAG0539) belonged to a member of the family Sandaracinaceae in the Myxococcota phylum. So far, there is only one cultured member of this bacterial family, *Sandaracinus amylolyticus* DSM 53668, a starch-degrading myxobacterium, also with a large genome (10.3 Mb)[66]. The degree of taxonomic novelty of the MDeep-MAGs associated with Myxococcota was particularly high, with four MAGs representing new potential genera. Therefore, we may have uncovered new niches in the bathypelagic ocean for such large-genome microbes, some of which have been shown to produce antibacterial, antifungal, and other bioactive metabolites[67].

**Genome-resolved metabolic capabilities of the bathypelagic ocean microbiome.** The MDeep-MAGs catalog represents the most extensive genome dataset from the bathypelagic ocean built to investigate the distribution and functional capability of deep-ocean microorganisms (Fig. 6). In the following section, we describe the most important features in the dataset.

*Non-cyanobacterial diazotrophs (NCDs).* The ecological relevance of non-cyanobacterial diazotrophs (NCDs) in the deep ocean remains unclear as the availability of carbon substrates seems

unlikely to support the costly energetic demands of $N_2$ fixation. However, NCDs are widely distributed across oxygenated oceans and are phylogenetically diverse[68–71]. New genotypes associated with Planctomycetota and Proteobacteria were recently found in the surface global ocean[62], and some of them were actively transcribed even at mesopelagic depths[37]. The diversity of NCDs in the deep ocean is mostly known from the detection of the *nifH* gene[72,73]. Three NCD MAGs were reconstructed in the Malaspina dataset that harbored the *nifH* gene and other structural genes of the *nif* operon such as *nifKD* (Supplementary Data 10): two were Alphaproteobacteria (MAG0509 and MAG0177) related to *Salipiger thiooxidans* and genus *Novosphingobium*, both with almost complete genomes (>94% completeness) that were detected exclusively in the PA fraction. The third one was a Gammaproteobacteria (MAG0081) in the genus *Ketobacter* present in both size fractions (Fig. 7). The presence of the Alphaproteobacteria diazotrophic MAGs in the PA fraction fits well with the finding of diazotrophic bacteria in sinking mesopelagic particles at the North Pacific Subtropical Gyre[74]. Nevertheless, the distribution of these three MAGs was restricted to a limited number of samples (between 5 and 9% of the total; Supplementary Data 10).

MAG0509 is closely related to *Salipiger thiooxidans*, a sulfur-oxidizing lithoheterotrophic bacterium isolated from the Black Sea[75]. Interestingly, in addition to the *nifH* and other structural genes from the *nif* operon (*nifK*), this MAG has two genes of the RuBisCo (forms I and IV), the *cox* genes for CO oxidation and a *soxY* gene for thiosulfate oxidation, pointing to potential chemolithoautotrophic diazotrophy (Supplementary Data 10).

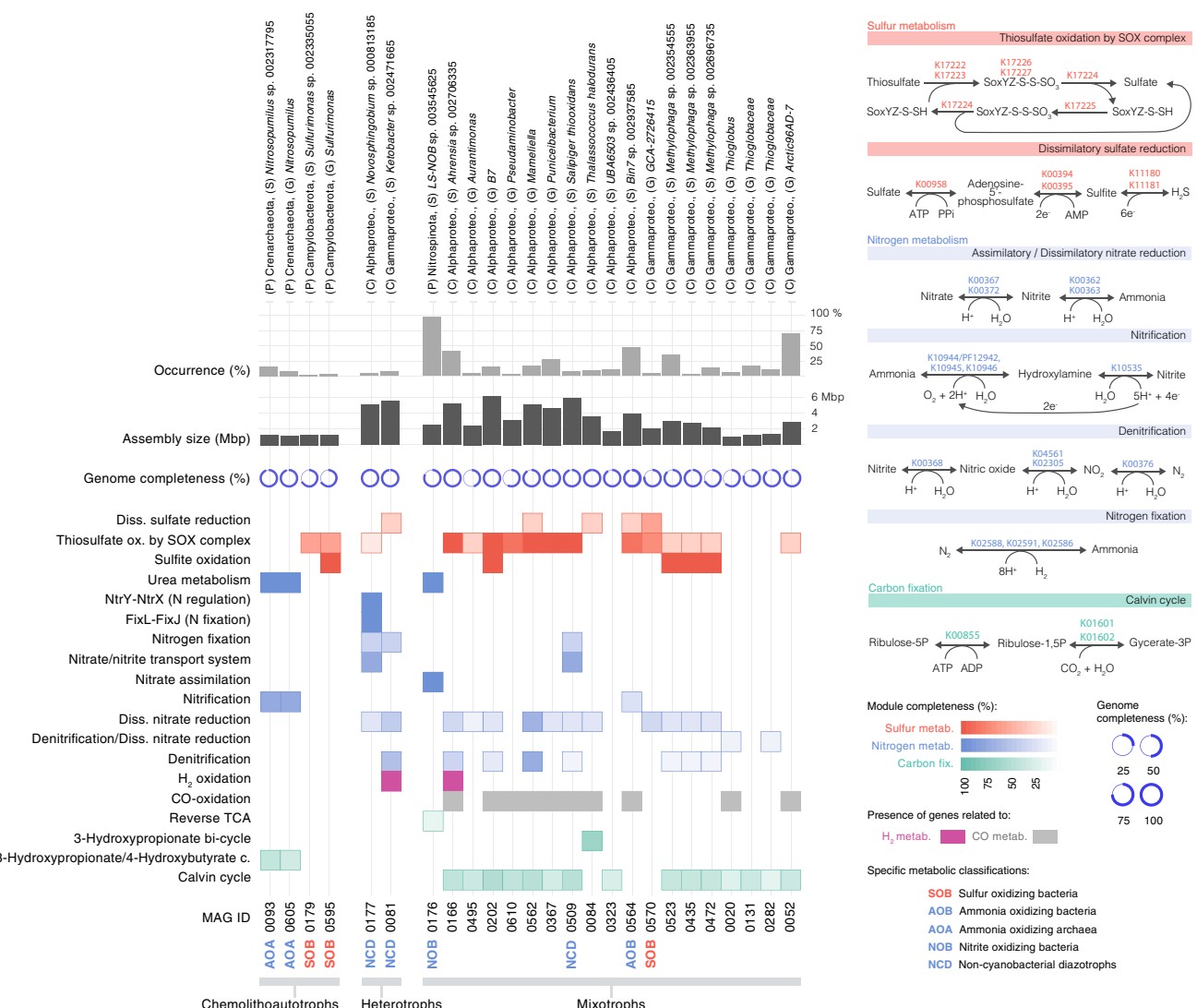

**Fig. 6 Metabolic potential in selected metagenome-assembled genomes (MAGs) from the bathypelagic ocean.** A total of 25 MDeep-MAGs were selected based on the presence of metabolic pathways with potential for chemolithoautotrophy, mixotrophy, and nitrogen fixation (non-cyanobacterial diazotrophs, NCDs). AOA ammonia-oxidizing archaea, SOB sulfur-oxidizing bacteria, NCDs non-cyanobacterial diazotrophs, NOB nitrite-oxidizing bacteria, AOB ammonia-oxidizing bacteria. Their taxonomic assignment at the maximal possible resolution is shown at the top, followed by the occurrence of each MAG in samples of the Malaspina Gene DataBase, and the genome completeness (%) of each MAG. On the right, metabolic pathways involved in inorganic carbon fixation (green), sulfur (red), nitrogen (blue) are shown as well as the KOs that participate in these pathways. At the bottom, the percentage module completeness for each pathway is coded by color intensity.

These results may reflect a higher metabolic versatility within this taxon, with the presence of previously undetected chemolithoautotrophic diazotrophs in the bathypelagic ocean. The other Alphaprobacteria genome related to *Novosphingobium* (MAG0177) was also intriguing: although members of this genus are known to be metabolically versatile and usually associated with the biodegradation of aromatic compounds, they have been commonly isolated from sites impacted by anthropogenic activities[76,77]. The only known species capable of $N_2$ fixing is *Novosphingobium nitrogenifigens*, isolated from a pulp and paper wastewater bioreactor with the ability to accumulate polyhydroxyalkanoate[78]. MAG0177 displayed a wide range of xenobiotic biodegradation pathways, such as those for xylene, toluene, and benzoate degradation (Supplementary Data 10).

Finally, the gammaproteobacterial NCD MAG was related to genus *Ketobacter* (MAG0081), with *Ketobacter alkanivorans* as the only species isolated from seawater impacted by an oil-spill accident, and with capacity for degradation of alkanes[79]. We observed that

alkane hydroxylase (*alkB*) and haloalkane dehalogenases (*dhaA*) genes were also detected in this genome (Supplementary Data 10) and therefore it may likely be capable of degrading synthetic haloalkanes, some of them used to make Nylon filament, fiber, and plastics and other recalcitrant compounds.

The phylogenetic analyses of these three *nifH* gene variants from our NCDs MAGs confirmed their relationship with Alphaproteobacteria and Gammaproteobacteria (Supplementary Fig. 6). MAG0081 clustered with the phylogenetic group I described by Delmont et al.[62] of Gammaproteobacteria and matched at 100% identity with other MAGs from the *Tara* Oceans expedition[62,80]. The other MAGs were related to the Alphaproteobacteria: the MAG0177 grouped with *Novosphingobium* sp. BW1 while the MAG0509 was related to *Yangia*, a genus of the Rhodobacteraceae.

The discovery of novel NCDs genomes from the bathypelagic ocean with presence in the PA fraction reinforces the idea of nutrient-rich sinking particles from the photic ocean as potential

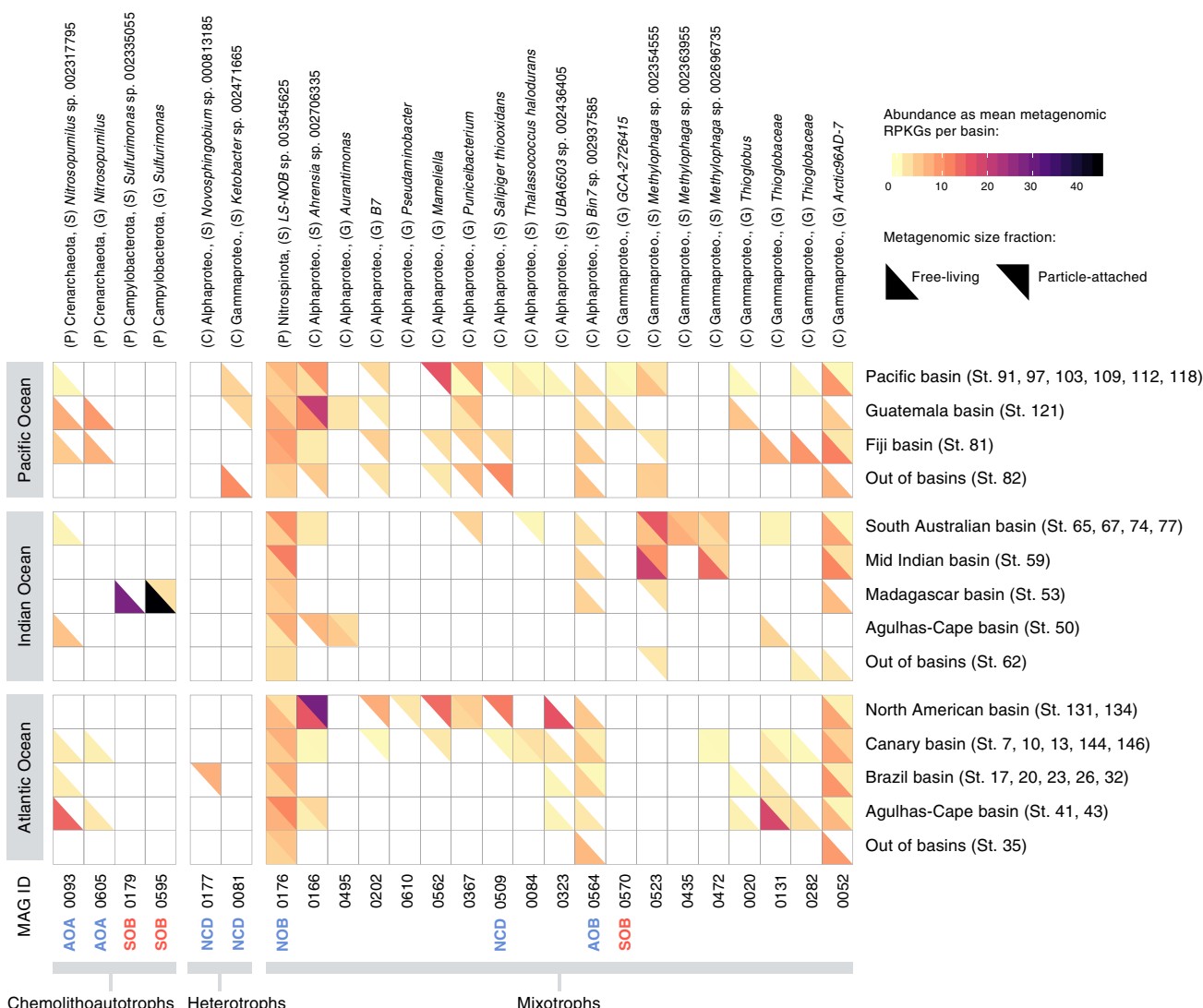

**Fig. 7 Mean abundance of 25 selected MAGs per oceanographic basin and size fraction, represented as metagenomic RPKGs (reads per genomic kilobase and metagenomic gigabase).** The upper half-tile represents RPKGs from the particle-attached size fraction metagenomes (0.8–20 μm) and the bottom half-tile represents RPKGs from the free-living size fraction metagenomes (0.2–0.8 μm). White represents the absence of the MAG in the metagenomic sample. MAGs are arranged based on their assigned metabolic strategy (chemolithoautotrophs, heterotrophs, and mixotrophs), and specific metabolic pathways confirmed in previous analyses are prepended to each MAG's ID following color codes from Fig. 6 (blue for nitrogen metabolism, red for sulfur metabolism; AOA ammonia-oxidizing archaea, SOB sulfur-oxidizing bacteria, NCDs non-cyanobacterial diazotrophs, NOB nitrite-oxidizing bacteria, AOB ammonia-oxidizing bacteria). Phylogenomic taxonomic assignation of MAGs is presented at the top of the figure.

niches for $N_2$ fixation, but further exploration would be needed. Our NCDs MAGs included previously undetected chemolithoautotrophic diazotrophs in the bathypelagic ocean. These nitrogen-fixing microorganisms appear metabolically diverse and coupled to multiple biogeochemical cycles.

*Chemolithoautotrophy and mixotrophy.* The ecological relevance of chemolithoautotrophy in the deep ocean is well known from contrasting oceanic regions, such as the North Atlantic mesopelagic and bathypelagic[8,81], the central Mediterranean Sea[82,83], or the Arctic Ocean[21,51]. Bacteria and archaea in the oxygenated water column of the dark ocean use a variety of reduced inorganic compounds, such as hydrogen, thiosulphate/sulfide, and ammonia, as energy sources[8,17,21,53]. However, it is unclear which are the key microbes involved and what is their prevalence in the global bathypelagic ocean. Therefore, we investigated the metabolisms associated with ammonia, sulfur, and nitrite oxidation that have been reported to be relevant in the bathypelagic

ocean[17,21,51] and other less explored metabolisms such as $H_2$-oxidation and CO oxidation. We found two ammonia-oxidizing archaea (AOA), three potential sulfur-oxidizing bacteria (SOB), one ammonia-oxidizing bacteria (AOB), and one nitrite-oxidizing bacterium (NOB) (Fig. 6).

The archaeal AOA (MAG0093, MAG0605) belonged to different species of *Nitrosopumilus*, found only in the free-living fraction (Fig. 7) and with an occurrence of 12% of the samples (Fig. 6 and Supplementary Data 10). Phylogenetic analyses of the *amoA* genes in the AOA MAGs showed that they were closely related to other deep-sea *amoA* sequences (Supplementary Fig. 5). In both AOA MAGs, we detected the key enzyme for the archaeal 3-hydroxypropionate–4-hydroxybutyrate autotrophic carbon dioxide assimilation pathway (K14534: *abfD*; 4-hydroxybutyryl-CoA dehydratase/vinylacetyl-CoA-delta-isomerase (Fig. 6). The two bacterial SOB (MAG0179, MAG0595) taxonomically associated with genus *Sulfurimonas*, displayed a narrow distribution at a single station (St53 in the Madagascar basin) and were enriched

in the free-living fraction while the distribution of the SOB related to Gammaproteobacteria GCA-2726415 (MAG0570) was limited to the Guatemala and Pacific basins in both size fractions (Fig. 7). However, we did not find in these SOB MAGs any key gene for inorganic carbon fixation, so their chemolithoautotrophy potential remains unknown. The potential AOB (MAG0564) related to MarineAlpha9-Bin7 harbored the key genes for nitrification (hao) and for CO oxidation (coxL; Fig. 6) and displayed a wider distribution (48% of the samples) across the Atlantic, Pacific, and Indian Oceans, mostly restricted to the free-living fraction (Fig. 7 and Supplementary Data 10). Finally, the bacterial NOB (MAG0176), related to the family Nitrospinaceae, was present in both size fractions in nearly all samples (Fig. 7).

The prevalence of the potential for the oxidation of CO by CO dehydrogenase (CODH; cox genes) in the bathypelagic ocean fits with the findings of a recent survey in which the coxL gene was widely distributed among aerobic bacteria and archaea in many terrestrial and marine environments[52]. We found a total of 90 (28%) MDeep-MAGs containing the coxL (K03520) gene, 46% of which also had the RuBisCo genes (large-chain rbcL) of the CBB cycle, supporting their potential for autotrophy. However, only two MAGs contained representative genes of $H_2$-oxidation (K00436: hoxH gene; Fig. 6): one Alphaproteobacteria of the genus Ahrensia (MAG0166) and one Gammaproteobacteria of the Ketobacter genus (MAG0081) that displayed different biogeography (Fig. 7). The Ahrensia MAG also contained the RuBisCo gene for inorganic carbon fixation and it was present in over 40% of the samples (Fig. 6 and Supplementary Data 10) with a peak in the North American basin (St131 and St134), being present in both size fractions, although more abundant in the PA microbial communities (Fig. 7).

The relevance of some of the inorganic carbon-fixation pathways, such as the CCB and the rTCA cycle associated with nitrite-oxidizing bacteria has been reported for the dark oceans, particularly for the mesopelagic and to a lesser extent in the bathypelagic ocean[18], by several studies combining single-cell genomics with fluorescence in situ hybridization and/or microautoradiography[17,18]. Yet, the prevalence of the different inorganic carbon-fixation pathways in the bathypelagic ocean at a broad geographical scale and at genome level was unknown.

The marker gene RuBisCo of the CCB cycle (K01601: large-chain rbcL) was found in 18 metagenomic bins, of which 15 are considered MAGs (totaling 4.7% out of the 317 MDeep-MAGs), whereas only MAG0084, an Alphaproteobacteria and MAG0176 belonging to Nitrospinota displayed marker genes related to alternative inorganic carbon-fixation pathways, such as the 3-HP and the rTCA cycle, respectively (Fig. 6). Two archaeal genomes (MAG0093 and MAG0605) related to Nitrosopumilus had the key gene of the hydroxypropionate–hydroxybutyrate cycle pathway for inorganic carbon fixation (K14534: abfD; 4-hydroxybutyryl-CoA dehydratase/vinylacetyl-CoA-delta-isomerase; Supplementary Data 10).

The presence of representative genes of the 3-HP such as the key enzyme malyl-CoA lyase (K08691: mcl)[84], the 2-methylfumaryl-CoA isomerase (K14470: mct) and 3-methylfumaryl-CoA hydratase (K09709: meh) together with the bicarbonate carboxylase enzymes (K01961: accC, acetyl-CoA carboxylase) was surprising since this pathway was assumed to be present only in Chloroflexaceae[84]. MAG0084, taxonomically related to Thalassococcus halodurans with 98% genome completeness, displayed 67% of the 3-hydroxypropionate pathway KOs (13 out of 20), although their role in autotrophy remains uncertain. This genome was present in ~10% of the samples (Fig. 6, Supplementary Data 10) in both the FL and PA fractions (Fig. 7). Individual genes of the 3-HP pathway have recently been found in the uncultured deep-ocean SAR202

clade genomes[85,86] related to Chloroflexi, although the role of the 3-hydroxypropionate bi-cycle in this clade has been linked to the assimilation of intermediate metabolites produced by the degradation of recalcitrant dissolved organic matter rather than $CO_2$ fixation[85].

The CBB cycle was the most abundant and prevalent inorganic carbon-fixation pathway in the bathypelagic ocean. Phylogenetic analyses showed that 13 out of the 18 RuBisCo sequences were associated with Form I ($n = 9$) and Form II ($n = 4$) (Supplementary Fig. 7). The remaining five sequences were related to Form IV that may be involved in methionine salvage, sulfur metabolism, and D-apiose catabolism[87,88]. Thus, most of our RuBisCo sequences (72%) are potentially involved in autotrophy (Form I and Form II) and were taxonomically assigned to Gammaproteobacteria and Alphaproteobacteria (Supplementary Fig. 7).

The 13 MAGs coding for RuBisCo (having either the large-chain rbcL; K01601 or small-chain rbcS: K01602) also had the phosphoribulokinase (prkA) gene (Supplementary Data 10). More importantly, most of them had a complete SOX system for thiosulphate oxidation, likely providing energy for $CO_2$ fixation (Fig. 6 and Supplementary Data 10). Nevertheless, the ten MAGs with Form I and II RuBisCo genes also possessed a large array of organic compound transporters (from 4 to 150) such as the ATP-binding cassette (ABC) transporters associated to up to 13 different COGs[17] (Supplementary Data 11), or from 47 to 497 genes[89](Supplementary Data 12) based on Pfams linked to the uptake of organic compounds such as sugars, amino acids, or peptides, suggesting a potential mixotrophic lifestyle for these lineages (Supplementary Data 13).

The most ubiquitous genomes with RuBisCo genes were MAG0052, related to the phylum SAR324 and genus Arctic96AD-7, which was detected in >70% of the samples (Fig. 6 and Supplementary Data 10) in both the FL and the PA fraction (Fig. 7), followed by MAG0166, an Alphaproteobacteria associated to Ahrensia sp002706335 that was present in 41% of the samples and MAG0523, related to Methylophaga, detected in 36% of the samples. The other genomes showed a rather limited distribution (Supplementary Data 10). Overall, MAGs containing RuBisCO Form I or II genes occurred on average in 22% of the samples, revealing that mixotrophy is a relatively common trait in the bathypelagic ocean.

Some of these chemolithoautotrophic/mixotrophic MAGs are within the top 50 most abundant MAGs of our dataset, representing abundant taxa of the bathypelagic deep ocean (Supplementary Fig. 8). The MAG0176 (NOB) was in position 25, the MAG0052 ranked 29, and the MAG0166 was the 42 most abundant MAG in the dataset (Supplementary Data 14). Our results differed from the MAGs reconstructed from particle-associated bacteria collected in sediment traps at 4000 m depth in the North Pacific Subtropical Gyre, in which key enzymes involving autotrophic pathways were not detected[31]. This points to a different taxonomic and functional composition of particles, as sediment traps tend to collect large sinking particles but might miss slow-sinking or buoyant particles[90].

Although we lack experimental evidence that these genomes can indeed perform inorganic carbon fixation, our results reveal their potential genetic capacity and motivate future experiments to characterize the ecological relevance of these novel mixotrophic, chemolithoautotrophic and diazotrophic lineages in the deep ocean. This study provides evidence for the distribution and biogeochemical potential of 317 genomes in the deep ocean and future studies integrating metatranscriptomics, metabolomics and single-cell function analyses, using e.g., NanoSIMS should help to bridge genomic presence and activity. Our findings, together with the enrichment of different metabolic pathways associated with

either FL or PA prokaryotes, expand our view of the metabolic seascape of the deep-ocean microbiome.

**Conclusions**. The global metagenomic assessment of the deep-ocean microbiome with the creation of the Malaspina Gene DataBase and Malaspina Deep MAGs catalog uncovers potentially novel orders and classes of deep-ocean microorganisms and describes potentially relevant biogeochemical processes of the deep-ocean microbiome in the tropical and subtropical bathypelagic oceans. Our results show metabolic differentiation reflected in contrasting functional gene repertoires, between the FL and the PA prokaryotic assemblages with also a certain degree of functional patchiness across oceans and basins. Our study also provides evidence for diverse metabolic strategies in the deep ocean. The widespread distribution of different autotrophic pathways in the deep ocean and the prevalence of alternative energy sources such as CO oxidation, sulfur oxidation, and $H_2$-oxidation, supports the role of a multitude of autotrophic processes in subsidizing the heterotrophic metabolism supported by export flux from the photic layer. These autotrophic processes depend on inorganic compounds photosynthetically reduced in the upper ocean and therefore do not constitute primary production in a strict sense. Yet, these autotrophic processes channel additional energetic resources from the upper ocean into the deep-oceanic microorganisms, allowing respiratory demands in the deep sea over those supported only by particulate organic fluxes from the photic layer.

## Methods

**Sample collection and DNA extraction**. A total of 58 water samples were taken during the Malaspina 2010 expedition (http://www.expedicionmalaspina.es) corresponding to 32 different sampling stations globally distributed across the world's tropical and subtropical oceans (Fig. 1a). We focused on the samples collected at the depth of 4000 m, although a few samples were taken at shallower depths, all within the bathypelagic realm (average depth: 3731 m ± 495; standard deviation). Two different size fractions were analyzed in each station representing the FL (0.2–0.8 μm) and PA (0.8–20 μm) prokaryotic communities[47,91,92]. While these two communities include prokaryotes, the PA assemblage also included microbial picoeukaryotes and their putative symbionts and the FL assemblage included some viruses[26,27].

For each sample, 120 l of seawater were sequentially filtered through a 200- and a 20-μm mesh to remove large plankton. Further filtering was done by pumping water serially through 142-mm polycarbonate membrane filters of 0.8-μm (Merk Millipore, Darmstadt, Germany, Isopore polycarbonate) and 0.2-μm (Merck Millipore, Express Plus) pore size with a peristaltic pump (Masterflex, EW-77410-10). The filters were then flash-frozen in liquid $N_2$ and stored at −80 °C until DNA extraction for whole community high-throughput shotgun sequencing. The filters for metagenomic sequencing were cut in small pieces with sterile razor blades and half of each filter was used for DNA extractions, which were performed using the standard phenol-chloroform protocol with slight modifications[27,93]. Details regarding the DNA extraction have been presented before[26].

**Library preparation and sequencing**. Plate-based DNA library preparation for Illumina sequencing was performed on the PerkinElmer Sciclone NGS robotic liquid handling system using Kapa Biosystems' library preparation kit. A total of 200 ng of sample DNA was sheared to 270 bp using a Covaris LE220 focused-ultrasonicator. The sheared DNA fragments were size selected by double-SPRI and then the selected fragments were end-repaired, A-tailed, and ligated with Illumina compatible sequencing adaptors from IDT containing a unique molecular index barcode for each sample library. The prepared libraries were then quantified using KAPA Biosystem's next-generation sequencing library qPCR kit and run on a Roche LightCycler 480 real-time PCR instrument. The quantified libraries were then multiplexed into pools of 8 or 12 libraries each, and the pool was then prepared for sequencing in the DOE's Joint Genome Institute (JGI) on the Illumina HiSeq2000 sequencing platform utilizing a TruSeq paired-end cluster kit, v3, following a 2 × 150 indexed run recipe, and Illumina's cBot instrument to generate a clustered flowcell for sequencing.

**Data acquisition**. The description and availability of the different datasets can be found in Supplementary Table 1. Our Malaspina bathypelagic microbial metagenomes were sequenced by the DOE´s Joint Genome Institute (JGI). Raw and clean sequences were therefore obtained from DOE's JGI Integrated Microbial Genomes and Microbiomes (IMG/MER), as well as several data analyzed

(Metagenome Annotation Standard Operating Procedure for IMG, Nov 2012) available as JGI proposal ID 300784. Functional abundance tables contained the number of reads in each sample for every functional category within four different functional annotations: Cluster of Orthologous Groups[45] (COG), KEGG orthologs[44] (KOs), Protein families[46] (Pfam), and Enzyme Commission classification (EC). In all cases, abundance tables were downloaded directly from the IMG repository (accession numbers in Supplementary Table 1) using the "estimated gene copies" option[94]. These tables represented the read counts for every annotated function in every sample coming from assembled gene data, taking into account the mean contig coverage and corrected by gene length.

Additional metadata were collected during the expedition including environmental variables (salinity, potential temperature, and oxygen concentration), sampling station coordinates (latitude, longitude, and depth), and auxiliary data for every sample (filter size, ocean basin, and water mass; Supplementary Data 1).

**Statistics and reproducibility**. For every functional abundance table, a sub-sampled equivalent table was constructed in order to avoid biases due to the varying sequencing depth between samples. Subsampling was performed by generating a randomly rarefied table without replacement from the original one with the "rrarefy" function in *vegan*[95] package within R software[96].

**Generation of the Malaspina Gene Database (M-GeneDB)**. All 3,872,410 predicted coding sequences larger than 100 bp from each assembled metagenome were pooled and clustered at 95% sequence similarity and 90% sequence overlap of the smaller sequence with cd-hit-est[97] v.4.6 using the following options: -c 0.95 -T 0 -M 0 -G 0 -aS 0.9 -g 1 -r 1 -d 0 to obtain 1,115,269 non-redundant gene clusters (from now on referred simply as genes). These gene clusters were aligned to UniRef100[98] (release 2019-10-16) with diamond blastx[99] (v0.9.22; *e*-value 0.0001). The least common ancestor taxonomic assignation of UniRef100 best matches was obtained from NCBI's taxonomy database[100] (release 2020-01-30).

In order to explore the novelty of the M-GeneDB, we clustered it with the 46,775,154 non-redundant sequences from the *Tara* Oceans Microbial Reference Gene Catalog version 2 (OM-RGC.v2)[37] using cd-hit-est-2d[97] v.4.6 with the following options: -c 0.95 -T 48 -M 256000 -G 0 -aS 0.9 -g 1 -r 1 -d 0 to obtain a final catalog of 47,422,971 genes.

**Taxonomy of protist, prokaryotes, and viruses from metagenomic reads**
*Protist*. 18S miTags were extracted from the metagenomes and subsequently analyzed following Logares et al.[93]. These miTags were mapped at 97% similarity using Uclust[101] to the PR2 database[102] that was pre-clustered at 97% similarity (Supplementary Data 3).

*Bacteria and archaea*. Prokaryotic taxonomical tables were downloaded from IMG [Compare Genomes/Phylogenetic Dist./Metagenomes vs. Genomes] using the "estimated gene copies" option (i.e., estimated by multiplying by read depth when available instead of using raw gene count) and the 60+ Perc. Identity option. A phylum-level table was downloaded for all samples. For Proteobacteria, a class-level table was also downloaded and merged, composing a table with all phyla and Proteobacteria divided into their classes. Estimated gene copies for each sample were divided by the total copies in order to obtain relative abundances and correct for different sampling depths (Supplementary Table S4).

*Nucleocytoplasmic large DNA viruses (NCLDV)*. Nucleocytoplasmic large DNA viruses (NCLDV) marker genes, including major capsid proteins and DNA polymerases, were detected in the 58 Malaspina deep metagenomics samples with the use of previously described procedures[43] using NCVOG[103] and PSI-BLAST[104] (*e*-value <1e-3) (Supplementary Table S5). For the detection of virophage sequences, we first screened the metagenomic sequences by the proteome sequences of three virophages (Sputnik, Mavirus, OLV) using BLAST (*e*-value<0.001). Then the metagenomic hits were searched against UniRef100[98] using BLAST (*e*-value <1e-3). As a result, 365 metagenomic peptides had best hits to virophage sequences, of which 50 sequences exhibited >95% sequence identity to homologs from the Mavirus virophages infecting *Cafeteria roenbergensis*.

*Viral signal analysis*. The marker gene *terL* (large subunit of the terminase) was used to assess the diversity of bacterial and archaeal viruses in the deep-sea microbial metagenomes. *terL* genes were identified in the proteins predicted from the 58 metagenomes through hits to the PFAM domains PF04466 (terminase_3), PF03237 (terminase_6), PF03354 (terminase_1), and PF05876 (terminase_GpA). Genes shorter than 100 bp were discarded, leading to a dataset of 485 *terL* genes. First, a family-level affiliation (i.e., *Myoviridae*, *Podoviridae*, or *Siphoviridae*) of these sequences was obtained from a best blastp hit to the RefseqVirus database (threshold of 50 on bit score, 0.001 on *e*-value, and 50 on % of amino acid identity; Supplementary Table S6). For each sample, viral community composition was then calculated based on this family-level affiliation and the normalized coverage of the contig (i.e., contig coverage divided by contig length and sequencing depth of the sample, as in Brum et al.[105]). Next, these 485 "deep-sea" *terL* genes were clustered with all *terL* from the RefseqVirus database (*n* = 899, v72, 09-2015), from

"environmental phages" in Genbank ($n = 456$, downloaded on 07-2015), from the VirSorter Curated Dataset[106] ($n = 6600$), and from the Global Ocean Virome Dataset[106] ($n = 2674$, sequences from 91 Epi- and Mesopelagic viromes from *Tara* Oceans and Malaspina expedition) at 98% of nucleotide identity (threshold most consistent with genome-based population definition, i.e., ≥80% of genes shared at ≥95% average nucleotide identity, when tested on complete genomes from RefseqVirus), leading to 5701 OTUs. The 485 *terL* genes from Malaspina were distributed across 303 OTUs, including 300 unique to deep-sea samples (i.e., containing only Malaspina *terL* sequences).

**Marker enzymes from energy metabolisms**. Metabolisms with a key role in the main biogeochemical cycles in the deep ocean were studied with additional detail by choosing specific enzymes for nitrogen, sulfur, methane, hydrogen, and carbon-fixation metabolic pathways. Marker enzymes for different pathways were defined by exploring the corresponding Kyoto Encyclopedia of Genes and Genomes (KEGG)[44] pathway maps (KEGG release 76.0). Enzymes participating only in reactions within each map were defined as marker enzymes for this metabolic map. When possible, marker enzymes were assigned to a specific module within a map (e.g., enzymes only participating in denitrification module within the nitrogen metabolism map). Corrected abundance estimation for marker enzymes was obtained by dividing the number of reads from the EC functional abundance table (without subsampling) by the number of reads assigned to the prokaryotic single-copy gene *recA* (selected as COG0468). A table selection of 83 KOs was built (Supplementary Data 7) representing the main key marker genes for different metabolic pathways relevant in the deep ocean. Of those, a total of 49 KOs were found in the Malaspina Deep-Sea Gene Collection (59%; Supplementary Table 8).

**Nonmetric multidimensional scaling (NMDS) and permutational multivariate analysis of variance (PERMANOVA)**. Nonmetric multidimensional scaling (NMDS) with stable solution from random starts was performed for the ordination of samples based on functional similarity using the KO abundance tables (Fig. 2) and the Pfam, COG, and EC number abundance tables (Supplementary Fig. 4). The Bray–Curtis distance measure was used for the abundance tables. All NMDS analyses were performed with the subsampled version of each abundance table. The partitioning of the variance in the Bray–Curtis distance matrix among oceans and oceanic basins was performed using permutational multivariate analysis of variance (PERMANOVA)[107] through the *adonis* function in the *vegan* R package (v2.5.6)[95] with 10,000 permutations. The two samples corresponding to the shallowest sampling station (station 62; 2400 m depth) were excluded from Fig. 2 as appeared as clear outliers (i.e., showed the high distance to any other sample), but they were included in Supplementary Fig. 4.

**Metagenomic Assembled Genomes from Deep Malaspina bathypelagic samples**. To build the Malaspina Deep Metagenome-Assembled Genomes (MAGs) catalog (MDeep-MAGs), all 58 metagenomes from the Malaspina expedition were pooled and co-assembled (megahit v1.2.8; options:–presets meta-large–min-contig-len 2000)[108]. Resultant contigs were de-replicated with cd-hit-est (v4.8.1 compiled for long sequence support; MAX_SEQ = 10000000, with options -c 0.95 -n 10 -G 0 -aS 0.95 -d 0)[97]. With this procedure, we increased the sequence space and we obtained a total of 421,891 contigs larger than 2000 bp. Metagenomic reads were back-mapped to the contigs dataset (bowtie[109] v2.3.4.1 with default options), keeping only mapping hits with quality larger than 10 (samtools[110] v1.8). We then binned the contigs into a total of 619 bins according to differential coverage and tetranucleotide frequencies in metabat (v2.12.1; jgi_summarize_bam_contig_depths and metabat2 with default options)[111]. The second round of assembly was carried out within each bin with CAP3[112] (v2015-10-02; options -o 16 -p 95 -h 100 -f 9) to solve overlapping overhangs with 95% of sequence similarity between contigs. Assemblies from high-quality MAGs were examined visually in Geneious v10.2.4 and contigs that aligned fully to a larger contig were manually removed.

**Taxonomic annotation of Deep Malaspina metagenomic assembled genomes (MDeep-MAGs)**. MAGs' completeness and single-copy gene redundancy (contamination) were estimated in CheckM (v1.0.18, lineage_wf)[64], and the placement in the prokaryotic tree of life of each MAG with completeness larger than 50% and contamination lower than 10% was used to plot a tree depicting the phylogenetic relationships between them (Fig. 5; iTOL[113] v4). Finer taxonomic assignations of the resulting 317 MAGs were estimated against the Genome Taxonomy Database (release r89) using GTDB-Tk[114] (v1.0.2; classify_wf) (Supplementary Data 9). Briefly, the annotation relies on both taxonomic placement in a backbone tree (using marker genes) and average nucleotide identity (ANI) comparison based on ~150,000 genomes, spanning isolates, MAGs as well as SAGs.

**Deep Malaspina MAGs annotation and metabolic prediction**. All 619 bins were annotated, including gene prediction, tRNA, rRNA, and CRISPR detection with prokka[115] (v1.13) with default options, using the estimated Domain classification from CheckM output as an argument of the–kingdom option. Additionally, MAGs' predicted coding sequences were annotated against the KEGG orthology database[44]

with kofamscan[116] (v1.1.0; database timestamp 2019-10-15), and against the PFAM database (release 31.0) with hmmer[117] (v3.1b2) with options–domtblout -E 0.1.

Primarily KEGG orthology was used to determine the energetic metabolism of the Malaspina Deep-Sea MAGs. A total of 83 marker genes from carbon fixation, methane, nitrogen, hydrogen, and sulfur metabolisms were selected (Supplementary Data 7) and their presence was explored along the low-quality (LQ) bins and medium quality (MQ) and high-quality (HQ) MAGs (https://malaspina-public.gitlab.io/malaspina-deep-ocean-microbiome/). A selected pool of 25 MAGs highlighting the potential for chemolithoautotrophy, mixotrophy, and non-cyanobacterial diazotrophs (NCDs) (Supplementary Data 9and Supplementary Data 10) was further explored within the MDeep-MAGs.

**Estimation of module pathway completeness within the Deep Malaspina MAGs dataset**. Module completeness per MAG was estimated by calculating the percentage of KOs belonging to each module over its total KO number (KEGG release 2019-02-11).

**Deep Malaspina MAGs abundance in the global bathypelagic Ocean**. The abundance of each MAG was assessed by mapping competitively the reads from the 58 metagenomes against the MAGs contig database using blastn[104] (v2.7.1; options -perc_identity 70 -e-value 0.0001). Metagenomic reads were randomly subsampled to the smallest sequencing depth value (4,175,346 read pairs) with bbtools (v38.08, reformat.sh; https://sourceforge.net/projects/bbmap/)[118]. Only reads with alignment coverage larger than 90% were kept for downstream analyses. Likewise, we kept only those metagenomic reads with sequence identity higher than 95%. In these cases, the metagenomic read was assumed to belong to the reference bin[119]. In addition, we discarded reads mapping any region annotated as rRNA to avoid spurious hits to highly conserved regions. The abundance of each MAG was expressed as the number of mapped reads per genomic kilobase and sample gigabase.

**Phylogenetic analyses of Rubisco, *amoA*, and *nifH* gene markers**. The 18 RuBisCo large-chain (K01601) amino acid sequences from the 619 Deep Malaspina bins were aligned using Clustal Omega[120] v1.2.3 (default options and 100 iterations) against the RuBisCo large-chain reference alignment profile published by Jaffe et al.[121], together with the large-chain sequences from heterotrophic marine Thaumarchaeota published by Aylward and Santoro[122]. Maximum-Likelihood phylogenetic reconstruction was done with FastTree[123] v2.1.11 (default options) using Jones–Taylor–Thorton model. Phylogenetic tree editing was done in iTol[113] v4.

For ammonia-oxidizing (*amoA*) gene phylogeny, we downloaded all nucleotide sequences annotated as *amoA* for phylum Nitrospirae and classes Beta and Gammaproteobacteria and Nitrososphaeria, and all sequences annotated as *pmoA* for Archaea, Candidate division NC10, and classes Alpha and Gammaproteobacteria from NCBI (April 2020). We also included sequences of *amoA* of both marine epipelagic and deep pelagic Archaea from Alves et al.[124] (clades NP-Alpha-2.2.2.1, NP-Epsilon-2 and NP-Gamma-2.1.3.1) to a total of 1651 sequences. We translated the sequences to amino acids (bacterial genetic code) and we removed redundancy by clustering all amino acid sequences to 98% of identity with cd-hit[97] (v4.8.1) to a final dataset of 586 sequences. We then added 19 sequences from 17 MAGs annotated with either K10944 or PF12942 and aligned all of them in mafft[125] (v7.402) with the G-INS-i option. We built a maximum-likelihood tree with FastTree[123] (v2.1.10) with default options and the Jones–Taylor–Thorton maximum-likelihood model and plotted it in iTol[113] v4.

To analyze the phylogeny of the *nifH* gene of 3 potentially diazotrophic MAGs, we retrieved their blastp best hits against NCBI's nr database (July 2020). We then added all *nifH* sequences between 240 and 320 residues long belonging to phylum Proteobacteria at NCBI Identical Protein Groups (IPG) database (query ((nifH AND ("240"[SLEN]:"320"[SLEN]))) AND proteobacteria[Organism]) and, additionally, we included 9 *nifH* sequences from MAGs from the *Tara* Oceans expedition[62], making a total of 192 sequences. Alignment, tree building, and plotting were done as above for *amoA* gene.

**Reporting summary**. Further information on research design is available in the Nature Research Reporting Summary linked to this article.

## Data availability

All data generated or analyzed during this study are included in this published article (and its supplementary information files). All raw sequences are publicly available at both DOE's JGI Integrated Microbial Genomes and Microbiomes (IMG/MER) and the European Nucleotide Archive (ENA). Individual metagenome assemblies, annotation files, and alignment files can be accessed at IMG/MER. All accession numbers are listed in Supplementary Data 1. The co-assembly for the MAG dataset construction can be found through ENA at https://www.ebi.ac.uk/ena with accession number PRJEB40454, the nucleotide sequence for each MAG and their annotation files can be found through BioStudies at https://www.ebi.ac.uk/biostudies with accession S-BSST457 and also in the companion website to this manuscript at https://malaspina-public.gitlab.io/malaspina-deep-ocean-microbiome/.

## Code availability

All software used in this work is publicly available distributed by their respective developers, and it is described in "Methods", including the versions and options used. Additional custom scripts to assign taxonomy to the M-geneDB genes and to filter and format FRA results are available through BioStudies at https://www.ebi.ac.uk/biostudies with accession S-BSST457.

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

## Acknowledgements

We thank the R/V Hespérides captain and crew, the chief scientists in Malaspina expedition legs, and all project participants for their help in making this project possible. This work was funded by the Spanish Ministry of Economy and Competitiveness (MINECO) through the Consolider-Ingenio program (Malaspina 2010 Expedition, ref. CSD2008-00077). The sequencing of 58 bathypelagic metagenomes was done by the U.S. Department of Energy Joint Genome Institute, supported by the Office of Science of the U.S. Department of Energy under Contract No. DE-AC02 05CH11231 to SGA (CSP 612 "Microbial metagenomics and transcriptomics from a global deep-ocean expedition"). Additional funding was provided by the project MAGGY (CTM2017-87736-R) to S.G.A. from the Spanish Ministry of Economy and Competitiveness, Grup de Recerca 2017SGR/1568 from Generalitat de Catalunya, and King Abdullah University of Science and Technology (KAUST) under contract OSR #3362 and by funding of the EMFF Program of the European Union (MERCLUB project, Grant Agreement 863584). The ICM researchers have had the institutional support of the "Severo Ochoa Centre of Excellence" accreditation (CEX2019-000928-S). High-Performance computing analyses were run at the Marine Bioinformatics Service (MARBITS, https://marbits.icm.csic.es) of the Institut de Ciències del Mar (ICM-CSIC), Barcelona, Supercomputing Center (Grant BCV-2013-2-0001) and KAUST's Ibex HPC. We thank Shook Studio for assistance with figure design and implementation and the anonymous reviewers for their comments to improve this manuscript. This paper is dedicated to the memory of Professor Vladimir B. Bajic, 2019.

## Author contributions

S.G.A. conceived this research. The primary analyses of the data were performed by P.S., G.S., F.M.C.C., and S.G.A. Authors that analyzed specific data and/or contributed to the interpretation of findings were M.S., R.L., L.P., S.S., P.H., H.O., G.L.M., S.R., J.M.G., J.M.A., C.B. J.R., S.P., P.B., D.V., M.B.S., R.M., C.P.A., and C.M.D. M.R.L. also contributed to data visualization. I.S.A., A.K., S.A., T.G., V.B.B., and C.M.D. provided computational assistance and/or funding resources. C.M.D. was the chief coordinator of the Malaspina Expedition and J.M.G was the coordinator responsible for the collection of samples for this study. C.P.A., C.M.D., and J.M.G. contributed specially to improve the manuscript. S.G.A. wrote the paper, all coauthors revised and approved the submission.

## Competing interests

The authors declare no competing interests.

## Additional information

[1]Department of Marine Biology and Oceanography, Institute of Marine Sciences (ICM), CSIC, Barcelona, Spain. [2]Department of Biology, Institute of Microbiology and Swiss Institute of Bioinformatics, ETH Zurich, Zurich, Switzerland. [3]Department of Ocean Sciences, University of California, Santa Cruz, CA, USA. [4]Instituto de Oceanografía y Cambio Global, IOCAG, Universidad de Las Palmas de Gran Canaria, ULPGC, Gran Canaria, Spain. [5]Aix Marseille Univ., Université de Toulon, CNRS, Marseille, France. [6]Institute for Chemical Research, Kyoto University, Gokasho, Uji, Japan . [7]Cellular and Molecular Microbiology, Faculté des Sciences, Université libre de Bruxelles (ULB), Brussels, Belgium. [8]Interuniversity Institute for Bioinformatics in Brussels, ULB-VUB, Brussels, Belgium. [9]Department of Microbiology, The Ohio State University, Columbus, OH, USA. [10]Department of Microbiology, University of La Laguna, La Laguna, Spain. [11]Spanish Institute of Oceanography (IEO), Oceanographic Center of The Canary Islands, Dársena Pesquera, Santa Cruz de Tenerife, Spain. [12]King Abdullah University of Science and Technology (KAUST), Computational Bioscience Research Center (CBRC), Thuwal, Saudi Arabia. [13]Institut de Biologie de l'ENS (IBENS), Département de biologie, École normale supérieure, CNRS, INSERM, Université PSL, Paris, France. [14]Research Federation for the study of Global Ocean Systems Ecology and Evolution, Paris, France. [15]Department of Microbiology and Immunology, Rega Institute, KU Leuven – University of Leuven, Leuven, Belgium. [16]VIB Center for Microbiology, Leuven, Belgium. [17]European Molecular Biology Laboratory, European Bioinformatics Institute (EMBL-EBI), Wellcome Genome Campus, Hinxton, Cambridge, United Kingdom. [18]PANGAEA, Data Publisher for Earth and Environmental Science, University of Bremen, Bremen, Germany. [19]Structural and Computational Biology, European Molecular Biology Laboratory, Heidelberg, Germany. [20]King Abdullah University of Science and Technology (KAUST), Red Sea Research Center (RSRC), Thuwal, Saudi Arabia. [21]Department of Microbiology and Civil Environmental and Geodetic Engineering, The Ohio State University, Columbus, OH, USA. [22]Department of Systems Biology, Centro Nacional de Biotecnología (CNB), CSIC, Madrid, Spain. [23]King Abdullah University of Science and Technology (KAUST), Red Sea Research Center (RSRC) and Computational Bioscience Research Center (CBRC), Thuwal, Saudi Arabia. [24]Centre for Marine Ecosystems Research, School of Sciences, Edith Cowan University, Joondalup, WA, Australia. [25]Present address: U.S. Department of Energy Joint Genome Institute, Berkeley, CA, USA. [26]These authors contributed equally: Silvia G. Acinas, Pablo Sánchez, Guillem Salazar, Francisco M. Cornejo-Castillo. ✉email: sacinas@icm.csic.es

