## [Peer Review File · Communications Biology]

This manuscript has been previously reviewed at another Nature Research journal. This document only contains reviewer comments and rebuttal letters for versions considered at Communications Biology.

Reviewers' comments:

Reviewer #2 (Remarks to the Author):

Acinas et al. present the first large scale bathypelagic microbial metagenomic dataset. As a member of the field, I am excited to see such a dataset and believe it will have a profound impact on our ability to assess large-scale questions about microbial oceanography. I am happy to see that many of the comments from the last review have been effectively incorporated into this version of the manuscript. The order of magnitude increase in MAGs is superb and exactly what I would expect by the change in approach. As noted in the first review, many of the awkward sentences and word choice has been cleared up, but there are still some sections where the clarity can be improved, and word choice modified – noted below. I still have a few major sticking points with regards to the methodology and interpretation of the results that will likely require the authors to re-examine specific elements and claims. And there are several broad choices related to how the data is presented and the corresponding analysis that I think can further be improved by the authors that would help solidify the impact of the work.

Major concerns.

1. I thought that Reviewer 1 had a very interesting question in the last round of reviews. Commenting on finding that when compared to the epipelagic, the M-GeneDB had ~71% unique protein coding sequences, and Reviewer 1 raised the hypothetical “what if more of the surface ocean was added?”. As the authors point out, the latest version of the OMRGC.v2 recently had 41 polar samples and when repeated the value of unique protein coding sequences dropped to 58%. For the authors, I think this illustrates the bathypelagic is under sampled, but also that much of the ocean remains under sampled. I would like the authors to emphasize this comparison is a snapshot of the current known databases and many be a definitive example of epipelagic vs bathypelagic in the distant future. I would hate for future work to be judged as only surface or only bathypelagic because of this comparison.

2. Much of the gene-centric analysis is based on 4 table downloaded from IMG and a subsampling approach that uses the ‘rrarefy’ package in R. In looking at the description of this tool, I am not convinced this the correct package to use to perform this analysis. I am not convinced it is incorrect either. Just that the tool is built from community abundance data and there is a lack of assumptions from the package author and the manuscript authors on how this may impact the analysis. I would assume the most robust way to approach this would be to perform multiple subsamples in an effort to bootstrap the results to ensure they are robust. There also is a profound limitation in using the IMG pipeline for this analysis, something that is rare in many of the high impact papers this manuscript cites. The pipeline is predominantly based on a procedure developed prior to 2012 (as noted by the authors) and even if elements have been updated since, there are still many modern approaches, tools, and databases that may provide more accurate results.

3. When describing the individual MAGs (between Ln. 319-368) the authors rely heavily on the reported functionality of the nearest cultured representatives but fail to then take the next step and analyze the related MAG for that functionality. Case in point, Ln. 333 why not perform ANTISMASH analysis to examine extent of secondary metabolites in the MAG0539? Or for MAG0177, why not examine the annotations for the degradation of aromatic compounds? If not present, that seems to be relevant statement, more so than a relative that has the function.

4. The support for mixotrophy amongst the RuBisCo containing MAGs seems to be predicated on the presence of transporter subunits (ref. Table S10). In examination of the table, the Pfam HMM that has the most occurrence in the genomes is PF00005 “ATP-binding domain”. At this resolution, for this particular subunit, it is not possible to determine what the substrate is for the associated transporter. The ATP-binding domain is common for all ABC-type transporters, including inorganic nutrients, etc.

Many of the other detected Pfam HMMs are non-specific. The few Pfam HMMs with substrate specificity for organic compounds include lipoprotein, which may be used for internal cell transport, and glycine betaine, which could be an osmolyte or carbon source, but additional functional predictions would be required (see: doi.org/10.1038/nmicrobiol.2016.146).

Further, Figure 5 is misleading. For the chemolithoautotrophs, specifically the AOA, why is the 3HP/4HB pathway not on Figure 5? This is discussed in the paper, but the result is not displayed. Further, I could find no mention in the manuscript for the autotrophic metabolisms related to MAGs 0179, 0595, 0564, and 0570 and none of them have pathways denoted in Figure 5. This makes the chemolithoautotrophy claim difficult to interpret.

Clarifications.

Ln 174. "Thaumarchaeota, accounted for 21% of all sequences". This will be confusing to readers as the authors refer to the taxonomy used in previous citations, however, use a taxonomy from GTDB that replaces the Thaumarchaeota as Crenarchaeota.

Ln 177-185. It is unclear what is added to the manuscript with the inclusion of this section. It fails to support the difference between PA vs FL. This could be moved to a supplemental Results and Discussion section.

Ln 217-219. Some of the KO number have gene abbreviations, others do not. Please unify.

Ln 220-224 & 267-268. For both of these pathways, statistics should be provided in support of the similarity, as is shown for CO-oxidation and pmoA-amoA. For C fixation, a high p-value that suggests that PA vs FL are not different is missing. And a p-value for anaerobic metabolisms enrich in the PA community.

Ln 233-234. This difference between nitrate reductase/nitrite oxidoreductase can and needs to be resolved through a phylogenetic tree, similar to pmoA/amoA and rbcL.

Ln. 241-242. Does it reflect differences between mesopelagic and bathypelagic OR between oxic and anoxic?

Ln 285. There is no methodology to suggest how the MAGs were manually curated.

Ln 294-299. It should be mentioned here that this is based on the GTDB taxonomic schema. And instead of terms like "could be" the authors should say something like "MAG0213 was assigned to a novel Class".

Ln 305-310. It is unclear what "lowering the mapping efficiency" means. In the prior sentence, the authors refer to the size of the assemblies, showing a difference between FL and PA. But Table S1 and the methods lack further development of "mapping efficiency". Mapping does not appear to have been performed against the assemblies, only the MAGs. It is unclear how the authors move transition from one element to the other.

Ln 311-312. Other Tara Oceans MAGs have higher recruitment rates. In Tully et al 2017, the average rates were ~15% for all size fractions, with a wide range.

Ln 387-388. "the potential for using nitrogen, sulfur, and carbon as terminal electron acceptors". This is unclear. Is sulfur noted because MAG0509 is related to an organism called *S. thiooxidans*? Sulfur was not mentioned in the overview of relevant metabolisms in this section. Also, N₂ and CO₂ are not terminal electron acceptors in their respective fixation processes. They are both reduced by cellular terminal electron acceptors, like NADP.

Typos/Syntax/Word choice.

Ln 64. "The Calvin Benson-Bassham cycle was the most prevalent"

Ln 66. "While CO-oxidation or ammonia oxidation was enriched in the free-living" What? MAGs? Community?

Ln 101. "unknow" change to "unknown"

Ln 131. "build up" change to "constructed"

Ln 139. "gene redundancy" to "sequence redundancy"

Ln 151. "in a suite" is incorrect word choice

Ln 151-160. Overall this section is convoluted and confusing.

"On average 61% (\pm 14%, SD) of the predicted genes in each sample were found in exclusively the bathypelagic, which highlights the unique gene content of this dataset (Fig. 1C, Table S1). Each

sample contained $14 \pm 9\%$ of the predicted genes not found in any other Malaspina samples (Fig. 1C, Table S1). Station St62, in the Indian Ocean sampled at 2,400 m, showed the highest fraction of sample-specific genes with 43% of the total and it was also different in terms of taxonomic community composition (Fig. S2). This sample, together with four other stations located in the Brazil (St32), North Atlantic American (St134) and Guatemala basins (St121) harbored more than 30% of sample-specific genes, were all from the particle-attached size fraction and associated with circumpolar deep water and North Atlantic Deep Water masses (Table S1)."

Ln 172. "the main difference with the photic layers" change to "the main difference from the photic layer"

Ln 173. "remarkable" is incorrect word choice

Ln 183. "targeting bacteria" change to "known to target bacteria"

Ln 198. "All of"

Ln 200. "was still" change to "has been"

Ln 202. "at the bathypelagic deep ocean is the community's lifestyle" change to "in the bathypelagic deep ocean is community lifestyle"

Ln 203-204. "rather than their origin, despite a patchy geographic distribution and differences among the different oceanic basins". This sentence is poorly structured.

Ln 250. "it is carried out by" change to "has been associated with the"

Ln 253. "remarkable" is incorrect word choice

Ln 263. "were present to different extents" is awkward word choice.

Ln 270. Add "between 0.7-8% of microbial cells"

Ln 274. Change to "conversely DNRA was enriched in the"

Ln 291. "MDeep-MAGs"

Ln 301. "only around" change to "approximately"

Ln 305. Remove "in fact"

Ln 316. Remove "at least"

Ln 320. "0.3 Mbp up to almost 10.9 Mbp" change to "0.3 Mbp to 10.9 Mbp"

Ln 361-362. Awkward sentence structure.

Ln 392. Change "or" to "and"

Ln 396-398. This sentence repeats all of the details of the sentence in Ln 393-394.

Ln 402. Change "of" to "in".

Ln 445. Change "about" to "approximately".

Ln 454. Change to "n = 9" and "n = 4"

Ln 467. Change "that was" to "which was"

Ln 472. "Prevalent" seems like a strong word choice for something that is in 20% of samples.

Ln 475. Remove "out"

Ln 476. "different origin" is incorrect word choice

Ln 501. "in a strict sense"

Reviewer #3 (Remarks to the Author):

This revised resubmission of "Metabolic Architecture of the Deep Ocean Microbiome" rebuts previous reviewer's comments and presents a revision the manuscript as an appeal to the Editors. I have read carefully through the original reviewer's comments, and the authors rebuttal to reviewer comments and revised manuscript. My comments on this revision follow below.

I tried very hard to find the true novelty and significance in this revision, and in the veracity of rebuttal, but remained unconvinced with regard to some of the central claims of the manuscript.

One concern starts with the title itself, "Metabolic Architecture of the Deep Ocean Microbiome" a notion that is further elaborated on in the manuscript. One might presume from this title, and the claims of the manuscript, that the authors have gathered adequate spatial sampling of the deep ocean, so as to define its "metabolic architecture" as the title and manuscript imply. This is

misrepresentative, on two fronts: While the authors sampled ~30 stations (two size fractions at most), the majority of samples originated from just one abyssopelagic depth range (3000-4000m, mostly 4000 m), plus another 25 or so (TARA) samples from <1000 m depth, over several transects and ocean basins. Considering the enormity of the deep ocean, and that the sampling is skewed to one depth, this is a very very sparse sampling of the deep ocean microbiome. Certainly, the 317 MAGs presented cannot be comprehensively representative, even for a single sampling site, much less the entire deep ocean. This doesn't detract from the quality or utility or interesting features of these data – but certainly some of the global claims seem a bit over-reaching.

While not a particularly dense sampling given the depth and extend of the deep ocean, this survey probably does represent the largest sampling of ocean depths at 4000m so far, so it is reasonable to ask the question, "What new has been learned?" The main argument of this paper (from the title, and text) is that the "metabolic architecture" of the deep ocean was determined. My main concern with this is that mainly one depth range was studied (or two that are very different when TARA is included), and only a very few select marker genes in the whole metagenome were critically examined, to bolster support for the importance of chemoautotrophy. Chemoautotrophy in the deep ocean is not a new concept. In the case of the most abundant autotrophic marker gene the authors found (ribulose-bisphosphate carboxylase/ phosphoribulokinase, to represent the Calvin Benson Bassam cycle), these were found to occur in only 1.3% of the estimated cells. Considering this, and the fact that ribulose-bisphosphate carboxylase sometimes serves other biochemical functions, or may not be continually expressed, this is not a very compelling case. For other chemoautotrophic (nitrification, sulfur oxidation, CO oxidation), and anaerobic respiration (nitrate/nitrite reduction, sulfate reduction) pathway, these are already well known in marine pelagic habitats, and particularly on particles.

More specific comments follow:

Results

(lines 151-3) The authors state that "On average 61 (\pm 14%, SD) of the predicted genes in each sample were exclusively found in the present dataset, which highlights the unique gene content of the bathypelagic microbiome". Yet it has long been known that this was to be expected, since deeper sampling results in more genes detected. The authors own recent work recapitulates this well-known fact clearly. (Env. Micro. 2020, Sequencing Effort Dictates Gene Discovery in Marine Microbial Metagenomes. doi: 10.1111/1462-2920.15182.). The fact that more genes were detected with this deeper sampling was expected. The most interesting question here is: "What is the nature of the novel deep-sea genes, and how do they differ from genes found elsewhere?" Unfortunately, this manuscript does not address, nor answer, this fundamental question in a quantitative nor convincing way.

Taxonomic affiliations (lines 162-186)

This section is a bit of a recap of the authors previous work and results on the very same samples, and so are not really new, nor surprising or novel, see in: Molec Ecol. 2020. "Major imprint of surface plankton on deep ocean prokaryotic structure and activity", doi: 10.1111/mec.15454.; PNAS 2018 "Sinking particles promote vertical connectivity in the ocean microbiome" doi: 10.1073/pnas.1802470115.; ISME J. 2016, "Global diversity and biogeography of deep-sea pelagic prokaryotes". doi: 10.1038/ismej.2015.137.

This previously published work by these same authors makes this section at best confirmatory. Likewise, information on partitioning of taxa into "free-living" FL (0.2-0.8 um size fraction) or "particle-attached" (PA, 0.8-20um size fraction) fractions is a recapitulation of previous work published by these same authors. (ISME J. 2016, Large variability of bathypelagic microbial eukaryotic communities across the world's oceans, doi: 10.1038/ismej.2015.170.)

Functional architecture of the deep ocean microbiome (lines 187-278)

The beginning of this section focuses on the difference between large and small size fractions. Not surprisingly, the biggest differences were found not biogeographically, but between different size

fractions for taxa, and genes. This is an expected result, seen in other contexts and depths as well. This expected result does not really distinguish the deep ocean from other habitats in the water column, and in fact the authors have already published on it, with the very same samples. In their *Molec. Ecol.* 2015 paper, "Particle-association lifestyle is a phylogenetically conserved trait in bathypelagic prokaryotes" (doi: 10.1111/mec.13419), using the very same samples as in this *Nat. Comms* submission (from 2150-4000 m) they stated: "As a consequence, PA and FL communities had clear alpha- and beta-diversity differences that exceeded the global-scale geographical variation". So, the great differences between large vs. small size fraction results reported here, are nothing new. And as taxa vary, so do their genes, as expected.

Lines 208-212. The authors state "To explore the potential metabolic differences between FL and PA prokaryotic microbial communities (Fig.3 and Fig.S4), a selection of marker genes (Table S6) were searched in the bathypelagic metagenomic dataset and their abundances normalized using *recA*". They find that among these 49 KEGG orthologs are present.

The authors do not tell us however why these marker genes were selected? Are they the most abundant genes? Are they more abundant than genes expected for heterotrophy? Are they most representative of the community? Are they otherwise somehow statistically significant? Lacking further explanation, the reader is left to assume genes were selectively picked, to support the authors' presumptions. Inspection of Table S6 shows a somewhat biased and very specific selection of genes, many of them known to commonly occur in deep-sea bacteria or archaea (including most of those for chemolithotrophic energy generation, anaerobic respiration, or CO₂ fixation in Table S6.)

Lines 215-278. This section describes the (narrow) set of marker genes the authors chose to focus on, mostly in terms of their % representation in the large or small size fractions, or the different stations sampled. If chemoautotrophy is so prevalent, then how prevalent among the metagenome are these marker genes? We are not really told how abundant these marker genes are, relative to other relevant metabolic pathways, for example those involved in heterotrophy. Even if chemoautotrophic pathway genes are there, how frequently are they (or their products) being expressed and used? This in itself renders the argument for the general importance for chemoautotrophy in the deep ocean, somewhat moot. The idea of "mixotrophy" is discussed, but this concept, more accurately termed facultative chemoautotrophy, is well known, and should probably be better reference. Some genes, like Rubisco, are used for other purposes than CO₂ fixation as well, which the authors do admit.

Diversity and novelty of the Malaspina Deep MAGs catalogue. (lines 280-366)

These final sections discuss the analyses of 317 medium quality or better MAGs, assembled from the dataset. These data are certainly interesting, but they do not seem to provide much integrative or compelling conclusions broadly, with respect to the deep ocean microbiome.

Genome resolved metabolic capabilities of the bathypelagic ocean (lines 338-504)

A handful of the MAGs (25) are shown to contain some of the marker genes for nitrification, denitrification, nitrogen fixation, sulfur oxidation and so on – and this is discussed in some detail. But most of these genes and pathways are already known for deep-sea marine bacteria, with respect to the pathways, and many of the taxa that contain them. Again, these are interesting data, but not entirely compelling for making grand conclusions regarding the metabolic architecture of the deep-ocean microbiome.

In conclusion, this is a certainly a great dataset, and there are some really interesting findings, but the authors have taken a somewhat narrow approach in their analysis, with a somewhat biased focus on just a few hand selected pathways – which is not a broad description of "metabolic architecture". A large focus of the paper, the differentiation of large ("PA") and small ("FL") size fraction taxa and genes. Are there diverse taxa and metabolic strategies on particles, including chemoautotrophy? Yes, this is already well known, supported by this work. But it is not specific to the deep-sea, and has already been noted by these authors and others. Finally, the potential for chemolithotrophy (and anaerobic respiration, and nitrogen fixation) in deep-sea bacteria and archaea has been known for some time, and described in previously published works. One of the authors' central claims in the

abstract, that "Interestingly, the potential to grow both autotrophically and heterotrophically was a prevalent prokaryotic strategy..." simply is not well supported by the data and analyses. The most abundant carbon fixation marker gene, ribulose-bisphosphate carboxylase (which may be involved in processes other than autotrophy as well) was found in only ~1.3% of the cells. This does not make a very compelling case (along with simple thermodynamic considerations), for the centrality of chemolithotrophy that the authors put forward.

Point by point Letter manuscript COMMSBIO-20-2735-T.

Reviewer #2 (Remarks to the Author):

Acinas et al. present **the first large scale bathypelagic microbial metagenomic dataset**. As a member of the field, **I am excited to see such a dataset and believe it will have a profound impact on our ability to assess large-scale questions about microbial oceanography**. I am happy to see that many of the comments from the last review have been effectively incorporated into this version of the manuscript. The order of magnitude increase in MAGs is superb and exactly what I would expect by the change in approach. As noted in the first review, many of the awkward sentences and word choice has been cleared up, but there are still some sections where the clarity can be improved, and word choice modified – noted below. I still have a few major sticking points with regards to the methodology and interpretation of the results that will likely require the authors to re-examine specific elements and claims. And there are several broad choices related to how the data is presented and the corresponding analysis that I think can further be improved by the authors that would help solidify the impact of the work.

RESPONSE 1: We are very thankful to this reviewer for his positive feedback and further comments to improve the manuscript. It is important to remark that most of the current version of the paper is focused on the analyses of the Malaspina MAGs dataset. Figures 5, 6 and 7 and all the MAGs analyses were done using the most updated approaches suggested by reviewer 2, confirming the more relevant results that we obtained by gene-centric analyses, such as the presence of many RuBisCo containing MAGs among many other findings.

Concerns.

1. I thought that Reviewer 1 had a very interesting question in the last round of reviews. Commenting on finding that when compared to the epipelagic, the M-GeneDB had ~71% unique protein coding sequences, and Reviewer 1 raised the hypothetical “what if more of the surface ocean was added?”. As the authors point out, the latest version of the OMRGC.v2 recently had 41 polar samples and when repeated the value of unique protein coding sequences dropped to 58%. For the authors, **I think this illustrates the bathypelagic is under sampled, but also that much of the ocean remains under sampled. I would like the authors to emphasize this comparison is a snapshot of the current known databases and many be a definitive example of epipelagic vs bathypelagic in the distant future.** I would hate for future work to be judged as only surface or only bathypelagic because of this comparison.

RESPONSE 2: We are sorry that we failed to highlight these differences in the first version of the manuscript and we agree that most of the oceans are under-sampled and that this kind of surveys can be seen as a snapshot (we have stressed this point in the new version, lines 146-147), but also that certain ecosystems, such as the Arctic or the deep Ocean, harbour distinct communities that significantly contribute to the discovery of new functions and taxa, as we show for the bathypelagic ocean.

2. Much of the gene-centric analysis is **based on 4 tables downloaded from IMG and a subsampling approach that uses the 'rrarefy' package in R**. In looking at the description of this tool, **I am not convinced this the correct package to use to perform this analysis. I am not convinced it is incorrect either**. Just that the tool is built from community abundance data and there is a lack of assumptions from the package author and the manuscript authors on how this may impact the analysis. I would assume the most robust way to approach this would be to perform multiple subsamples in an effort to bootstrap the results to ensure they are robust. There also is a profound limitation in using the IMG pipeline for this analysis, something that is rare in many of the high impact papers this manuscript cites. The pipeline is predominantly based on a procedure developed prior to 2012 (as noted by the authors) and even if elements have been updated since, there are still many modern approaches, tools, and databases that may provide more accurate results.

RESPONSE 3: As the reviewer points out, the gene-centric analysis is performed based on a gene-abundance table that was down-sampled to correct for varying sequencing depth. This was done using the 'rrarefy' function in the 'vegan' R package. This function randomly rarefies the table so that a random sample of the counts is taken with the same sample size for all entries. This method has no specific assumption on the nature of the information stored in the abundance table. Thus, although implemented in vegan with species abundance in mind, it can perfectly be applied to any abundance data. Other implementations of the same approach, such as for example RTK (Saary et al, 2017), openly recognize the applicability of this approach for gene abundance data. Although a bootstrap approach, in which several independently downsampled tables are averaged, could also be used, as pointed out by the reviewer, this might also affect the variance of the gene abundances, especially for the low abundant ones. That's the reason we choose to use single downsampling.

Concerning the use of the IMG pipeline for gene annotation. The Integrated Microbial Genomes & Microbiomes (IMG/M) system supports the annotation, analysis and distribution of microbial genome and microbiome datasets sequenced at DOE's Joint Genome Institute (JGI) and their IMG pipelines and datasets are accurate despite the time passed. The data from IMG used for the gene-centric analysis was only used on selected gene marker genes, which are well- established genes. We believe that no further efforts at refining the annotation need to be performed for this specific analysis. It is also important to highlight that most of what work done in the present vs. has been dedicated to the reconstruction of metagenome-assembled genomes (MAGs), and most of the paper is devoted to the analyses of these MAGs.

Saary et al, 2017: <https://academic.oup.com/bioinformatics/article/33/16/2594/3111845>

3. When describing the individual MAGs (between Ln. 319-368) **the authors rely heavily on the reported functionality of the nearest cultured representatives** but fail to then take the next step and analyze the related MAG for that functionality. Case in point, Ln. 333 why not perform ANTISMASH analysis to examine extent of secondary metabolites in the MAG0539? **Or for MAG0177, why not examine the annotations for**

the degradation of aromatic compounds? If not present, that seems to be relevant statement, more so than a relative that has the function.

RESPONSE 4. We indeed related some of the MAG functionalities to the nearest cultured representative because we believe it is a relevant comparison since these isolates represent references with well know physiologies and accurate genome information. However, we have a full functional description of all MAGs at the KEGG level presented in the Supplementary tables that can be used to search for any function of interest. As suggested by this reviewer, we have complemented this paragraph with the examination of aromatic compounds degradation genes (such as xylene, toluene and benzoate degradation) in MAG0177 as an example (Lines 347-348), but for the sake of the length of this manuscript, we cannot describe all interesting functional traits arising from our data.

4. The support for mixotrophy amongst the RuBisCo containing MAGs seems to be predicated on the presence of transporter subunits (ref. Table S10). In examination of the table, the Pfam HMM that has the most occurrence in the genomes is PF00005 “ATP-binding domain”. At this resolution, for this particular subunit, it is not possible to determine what the substrate is for the associated transporter. The ATP-binding domain is common for all ABC-type transporters, including inorganic nutrients, etc. Many of the other detected Pfam HMMs are non-specific. **The few Pfam HMMs with substrate specificity for organic compounds include lipoprotein, which may be used for internal cell transport, and glycine betaine, which could be an osmolyte or carbon source,** but additional functional predictions would be required (see: doi.org/10.1038/nmicrobiol.2016.146).

RESPONSE 5: We searched for PF0005 along with other Pfam listed in Table S10 because some of these genes had been used in previously published studies for similar purposes (e.g. Swan et al 2011, or Yelton et al 2016) to explore mixotrophy within deep ocean lineages and picocyanobacterial genomes respectively. We agree that PF0005 may not be very resolute, and therefore, we have extended the search by using all COGs, Pfam, and EC numbers listed in Swan et al. 2011 and Yelton et al. 2016 within our potential mixotrophic MAGs listed in Fig. 6, thus providing a higher resolution to infer that many of these ABC transports are indeed for sugar, AA or oligopeptide transporters (see additional supplementary Tables for COGs, New **Table S11**, and Pfam, **Tables S12**). Additionally, we searched for the additional genes described in Daly et al. 2016 *Nat Microbiology*, as suggested by this reviewer, to extract extra information about the role of some of these ATP uptake systems for glycine betaine/proline and polar amino acids (**Table S13**). We have clarified the content in lines 452 to 455 to update this extra information and Table S11, Table S12 and Table S13 have been provided with the completed list of COGs, Pfams and a summary table respectively.

In summary, this additional search provides further support to our previous hypotheses on the mixotrophic capabilities by the listed potentially mixotroph MAGs listed in Fig. 5. The data appear solid but, obviously, experimental verification is missing.

References:

Swan, B. K. *et al.* Potential for chemolithoautotrophy among ubiquitous bacteria lineages in the dark ocean. *Science* **333**, 1296–1300 (2011).

Yelton, A. P. *et al.* Global genetic capacity for mixotrophy in marine picocyanobacteria. *ISME J.* 10, 2946–2957 (2016).

Daly *et al.* Microbial metabolisms in a 2.5-km-deep ecosystem created by hydraulic fracturing in shales. *Nat Microbiology.* 2016. DOI: 10.1038/NMICROBIOL.2016.146.

Further, Figure 5 is misleading. **For the chemolithoautotrophic, specifically the AOA, why is the 3HP/4HB pathway not on Figure 5?** The is discussed in the paper, but the result is not displayed.

RESPONSE 6: It is difficult to include all details in a single figure. Figure 5 (now it is Figure 6) does not aim to display all MAGs with autotrophic/chemolithoautotrophic metabolisms but rather highlights only 25 relevant MAGs (not all) with potential for chemolithoautotrophy, mixotrophy and non-cyanobacterial diazotrophs (NCDs). We present the rest of the information in supplementary tables as well as being presented in our companion website with all details, including the functional annotation for all genes for each MAG based on COGs, Pfams and KOs. However, based on the reviewer comment, we have included the key gene of the hydroxypropionate-hydroxybutyrate cycle pathway for inorganic carbon fixation (K14534: *abfD*; 4-hydroxybutyryl-CoA dehydratase / vinylacetyl-CoA-Delta-isomerase) labelled in green in the left section of Fig. 5. This gene was present in both Ammonia-Oxidizing Archaea (AOA) MAG0093 and MAG0605 (see new Figure 6 below).

Further, I could find no mention in the manuscript for the autotrophic metabolisms related to MAGs 0179, 0595, 0564, and 0570 and none of them have pathways denoted in Figure 5. This makes the chemolithoautotrophy claim difficult to interpret.

RESPONSE 7: The reviewer is correct and the genes for inorganic carbon fixation were not found in these partial MAGs related to the three potential SOB, and therefore we modified the text extending the description of these MAGs accordingly (lines 391-392). We now mention that the enzymes for inorganic carbon fixation were missing and therefore their potential chemolithoautotrophy in MAGs 0179, 0595 and 0570 remains elusive. The MAG0564 related to AOB contained the complete array of cox genes for CO-Oxidation, Because of their potential ecological relevance in the nitrification or sulfur oxidation, these four MAGs are presented in Figure 6.

Clarifications.

Ln 174. "Thaumarchaeota, accounted for 21% of all sequences". This will be confusing to readers as the authors refer to the taxonomy used in previous citations, however, use a taxonomy from GTDB that replaces the Thaumarchaeota as Crenarchaeota.

The reviewer is right and we have modified the text and use Crenarchaeota This section has been moved to supplementary results and discussion as requested by this reviewer.

Ln 177-185. It is unclear what is added to the manuscript with the inclusion of this section. It fails to support the difference between PA vs FL. This could be moved to a supplemental Results and Discussion section.

We have moved most of this section to supplemental Results and Discussion. We think a few sentences highlighting the main groups of protists, bacteria, archaea, giruses and viruses are always nice to read regardless that some of these findings are basically supportive of previous studies using amplicon Tags in protists and prokaryotes. Nevertheless, the results of giruses in the bathypelagic are first present here.

Ln 217-219. Some of the KO numbers have gene abbreviations, others do not. Please unify.

It is done for all except for K15039 for which we did not find any abbreviation for 3-hydroxypropionate dehydrogenase.

Ln 220-224 & 267-268. For both of these pathways, statistics should be provided in support of the similarity, as is shown for CO-oxidation and pmoA-amoA. For C fixation, a high p-value that suggests that PA vs FL are not different is missing. And a p-value for anaerobic metabolisms enrich in the PA community.

It has been updated now. For C fixation we added the Wilcoxon test p-value = 0.1347 in line 207-208 and for assimilatory and dissimilatory nitrate reduction pathways (Wilcoxon test p-value < 0.005) in line 251-252.

Ln 233-234. This difference between nitrate reductase/nitrite oxidoreductase can and needs to be resolved through a phylogenetic tree, similar to pmoA/amoA and rbcL.

We understand that the phylogeny would help to distinguish between nitrate reductase/nitrite oxidoreductase but taking into account that we have only reconstructed a single MAG related to nitrite-oxidizing bacteria (MAG0176) we believe this effort was not essential for the paper, since all related to NOB we mention in the ms is based on the NOB MAG0176.

Ln. 241-242. Does it reflect differences between mesopelagic and bathypelagic OR between oxic and anoxic?

The reviewer is right. It has been modified accordingly in lines 225-226 as follows: “This reflects different biogeochemical processes occurring in the anoxic mesopelagic OMZ and the oxic bathypelagic oceans.

Ln 285. There is no methodology to suggest how the MAGs were manually curated.

A new paragraph has been included in the supplementary material explaining the extra step we did to review the MAGs in which assemblies from high-quality MAGs were examined in Geneious v10.2.4 and contigs that aligned fully to a larger contig were manually removed (lines 692-694). However, we have changed the word "manually curated" from the main text since it probably cannot be considered as manual curation.

Ln 294-299. It should be mentioned here that this is based on the GTDB taxonomic schema. And instead of terms like “could be” the authors should say something like “MAG0213 was assigned to a novel Class”.

Done.

Ln 305-310. It is unclear what “lowering the mapping efficiency” means. In the prior sentence, the authors refer to the size of the assemblies, showing a difference between FL and PA. But Table S1 and the methods lack further development of “mapping efficiency”. Mapping does not appear to have been performed against the assemblies, only the MAGs. It is unclear how the authors move transition from one element to the other.

We have modified this sentence to clarify our results and it now reads (lines 286-291): One possibility is due to the presence of picoeukaryotes since protists, which have larger genomes, are more fragmented in the metagenome and are more difficult to bin into MAGs, as suggested by the lower individual assembly sizes of PA compared to FL (22.4 ± 13.6 and 36.9 ± 17.1 Mbp respectively; **Table S1**) obtained with the same sequencing depth. This lower read mapping coming from these large size fraction samples may be due to the lower genome reconstruction of picoeukaryotes

Ln 311-312. Other Tara Oceans MAGs have higher recruitment rates. In Tully et al 2017, the average rates were ~15% for all size fractions, with a wide range.

This is right and now we added a comment of this data of higher recruitment rates from Tully et al 2017 in the ms. in line 295-297.

Ln 387-388. “the potential for using nitrogen, sulfur, and carbon as terminal electron acceptors”. This is unclear. Is sulfur noted because MAG0509 is related to an organism called *S. thiooxidans*? Sulfur was not mentioned in the overview of relevant metabolisms in this section. Also, N₂ and CO₂ are not terminal electron acceptors in their respective fixation processes. They are both reduced by cellular terminal electron acceptors, like NADP.

We agree that this sentence was confusing and we have re-written this paragraph in lines 389-392.

Typos/Syntax/Word choice.

Ln 64. “The Calvin Benson-Bassham cycle was the most prevalent”
Corrected (line 61).

Ln 66. “While CO-oxidation or ammonia oxidation was enriched in the free-living”
What? MAGs? Community?
Corrected now it reads “Ammonia and CO oxidation pathways are enriched in the free-living microbial communities ” (line 58-59).

Ln 101. “unknow” change to “unknown”
Done.

Ln 131. “build up” change to “constructed”.
Done.

Ln 139. “gene redundancy” to “sequence redundancy”
Done.

Ln 151. “in a suite” is incorrect word choice
Corrected.

Ln 151-160. Overall this section is convoluted and confusing.
“On average 61% (\pm 14%, SD) of the predicted genes in each sample were found in exclusively the bathypelagic, which highlights the unique gene content of this dataset (Fig. 1C, Table S1). Each sample contained $14 \pm 9\%$ of the predicted genes not found in any other Malaspina samples (Fig. 1C, Table S1). Station St62, in the Indian Ocean sampled at 2,400 m, showed the highest fraction of sample-specific genes with 43% of the total and it was also different in terms of taxonomic community composition (Fig. S2). This sample, together with four other stations located in the Brazil (St32), North Atlantic American (St134) and Guatemala basins (St121) harbored more than 30% of sample-specific genes, were all from the particle-attached size fraction and associated with circumpolar deep water and North Atlantic Deep Water masses (Table S1).”
We have modified some of these sentences trying to make them more clear. Lines 151-160.

Ln 172. “the main difference with the photic layers” change to “the main difference from the photic layer”
Done.

Ln 173. “remarkable” is incorrect word choice
We have removed this word and changed it by “relevance of”

Ln 183. “targeting bacteria” change to “known to target bacteria”
Done

Ln 198. “All of”
Revised

Ln 200. “was still” change to “has been”
Revised.

Ln 202. “at the bathypelagic deep ocean is the community’s lifestyle” change to “in the bathypelagic deep ocean is community lifestyle”
Revised.

Ln 203-204. “rather than their origin, despite a patchy geographic distribution and differences among the different oceanic basins”. This sentence is poorly structured.
Thanks, it has been modified to: “rather than their geographic origin, although differences among the different oceanic” Line 188-189.

Ln 250. “it is carried out by” change to “has been associated with the”
Done.

Ln 253. “remarkable” is incorrect word choice
Changed by notable.

Ln 263. “were present to different extents” is awkward word choice.
We changed to “were also present”

Ln 270. Add “between 0.7-8% of microbial cells”.
Done.

Ln 274. Change to “conversely DNRA was enriched in the”
Done.

Ln 291. “MDeep-MAGs”
Done.

Ln 301. “only around” change to “approximately”
Done.

Ln 305. Remove “in fact”
Done.

Ln 316. Remove “at least”
Done.

Ln 320. “0.3 Mbp up to almost 10.9 Mbp” change to “0.3 Mbp to 10.9 Mbp”
Done.

Ln 361-362. Awkward sentence structure.
Revised.

Ln 392. Change “or” to “and”
Done.

Ln 396-398. This sentence repeats all of the details of the sentence in Ln 393-394.
We think it is OK.

Ln 402. Change “of” to “in”.
Done.

Ln 445. Change “about” to “approximately”.
Done.

Ln 454. Change to “n = 9” and “n = 4”
Done.

Ln 467. Change “that was” to “which was”
Done.

Ln 472. “Prevalent” seems like a strong word choice for something that is in 20% of samples.
Agree, the potential mixotrophy occurred on average in 22% (some of the RuBisCo MAGs are in 70 or 40% of the samples) and we modified to “relatively common”. Line 463-464.

Ln 475. Remove “out”
Done.

Ln 476. “different origin” is incorrect word choice
We have removed these words.

Ln 501. “in a strict sense”
Done.

RESPONSE 8: Thanks for this careful revision of our manuscript. All these typos and small suggestions have been corrected and text modified accordingly.

Reviewer #3 (Remarks to the Author):

This revised resubmission of “Metabolic Architecture of the Deep Ocean Microbiome” rebuts previous reviewer’s comments and presents a revision the manuscript as an appeal to the Editors. I have read carefully through the original reviewer’s comments, and the authors rebuttal to reviewer comments and revised manuscript. My comments on this revision follow below.

I tried very hard to find the true novelty and significance in this revision, and in the veracity of rebuttal, but remained unconvinced with regard to some of the central claims of the manuscript.

One concern starts with the title itself, “Metabolic Architecture of the Deep Ocean Microbiome” a notion that is further elaborated on in the manuscript. One might presume from this title, and the claims of the manuscript, that the authors have gathered adequate spatial sampling of the deep ocean, so as to define its “metabolic architecture” as the title and manuscript imply. This is misrepresentative, on two fronts: While the authors sampled ~30 stations (two size fractions at most), the majority of samples originated from just one abyssopelagic depth range (3000-4000m, mostly 4000 m), plus another 25 or so (TARA) samples from <1000 m depth, over several transects and ocean basins. **Considering the enormity of the deep ocean, and that the sampling is skewed to one depth, this is a very very sparse sampling of the deep ocean microbiome.** Certainly, the 317 MAGs presented cannot be comprehensively representative, even for a single sampling site, much less the entire deep ocean. This doesn’t detract from the quality or utility or interesting features of these data – but certainly some of the global claims seem a bit over-reaching.

RESPONSE 9: The reviewer considers that our sampling is not representative of the whole bathypelagic ocean, which might be true, yet he also claims that *“probably does represent the largest sampling of ocean depths at 4000m and 317 MAGs of medium-high quality would enrich significantly our acknowledgement of the functional/metabolic capacity of the bathypelagic deep ocean”*. Not a single study of the ocean or any ecosystem can be considered exhaustive, but we believe that our sampling covered reasonably well the diversity of the tropical and subtropical oceans at depths of ca. 4000 m and in two size classes. Our dataset is the largest ever published of the bathypelagic deep ocean. And this is recognized by the other reviewers. See e.g. the comment of **reviewer #2:** *“This study presents the first large scale bathypelagic microbial metagenomic dataset. As a member of the field, I am excited to see such a dataset and believe it will have a profound impact on our ability to assess large-scale questions about microbial oceanography”*. In any case, reviewer#3 himself states that *“317 MAGs of medium-high quality would enrich significantly our knowledge of the functional/metabolic capacity of the bathypelagic deep ocean”*. The journal has suggested an alternative title for this manuscript which would be: **“Deep ocean metagenomes provides insight into the metabolic architecture of bathypelagic microbial communities”**. I think with this revised title the reviewer will be more comfortable.

While not a particularly dense sampling given the depth and extend of the deep ocean, this survey **probably does represent the largest sampling of ocean depths at 4000m so far**, so it is reasonable to ask the question, “What new has been learned?” The main argument of this paper (from the title, and text) is that the “metabolic architecture” of the deep ocean was determined. My main concern with this is that mainly one depth range was studied (or two that are very different when TARA is included), and only a very few select marker genes in the whole metagenome were critically examined, to bolster support for the importance of chemoautotrophy. **Chemoautotrophy in the deep ocean is not a new concept. In the case of the most abundant autotrophic marker gene the authors found (ribulose-bisphosphate carboxylase/ phosphoribulokinase, to represent the Calvin Benson Bassam cycle), these were found to occur in only 1.3% of the estimated cells. Considering this, and the fact that ribulose-bisphosphate carboxylase sometimes serves other biochemical functions, or may not be continually expressed, this is not a very compelling case.** For other chemoautotrophic

(nitrification, sulfur oxidation, CO oxidation), and anaerobic respiration (nitrate/nitrite reduction, sulfate reduction) pathway, these are already well known in marine pelagic habitats, and particularly on particles.

RESPONSE 10: The reviewer makes some considerations about the generality/relevance of our results (which we answered above). He/she then claims that our results do not demonstrate that the metabolic architecture of the deep ocean includes a large number of chemolithoautotrophic organisms. The reviewer bases this comment on the average relative abundance of 1.3% of the RuBisCo genes. Yet this is “on average”, while in some of the stations, such as in St134, it could reach 12% of the cells. These values are indeed a significant fraction of the total bacterioplankton and likely more when one refers to a single genome/lineage (MAGs). Most importantly, what we have stressed in our paper is not the abundance of the RuBisCo gene (not only RuBisCo but also other genes for different autotrophic pathways) but rather its prevalence in the bathypelagic deep ocean based on our MAGs reconstruction. The genomes (the MAGs) containing the RuBisCo gene Form I and II, which are considered truly linked to autotrophy, were present in about 22% of the bathypelagic samples on average. Also, most of these MAGs had a complete SOX system for thiosulphate oxidation, likely providing energy for CO₂ fixation (**Fig.6; Table S10**), so we believe that those genomes have the real potential for **chemoautotrophy**. Thus, some of these RuBisCo -containing MAGs, such as MAG0052 related to the phylum SAR324 and genus Arctic96AD-7 were detected in > 70% of the samples (**Fig. 6, Table S10**) in both the free-living and the particle-attached fractions. Although indeed RuBisCo -containing SAGs have already been described (Swan et al. 2011), our study was designed to assess the prevalence of this pathway at a global scale. We also uncovered new lineages that had not been described before, such as **MAG0509** who represent the **first diazotrophic autotrophic genome detected in the bathypelagic ocean** (contains the *RuBisCo* and the PRK genes of the Calvin Cycle, the complete SOX complex for Thiosulfate oxidation and *nifH* genes), and we thus believe that such information is relevant and novel. Similarly, although other chemoautotrophic pathways have been already described in local studies, our work provides the opportunity to assess the presence and prevalence of these pathways in the largest sampling effort of 4000 m depth waters performed to date (and here we are quoting the reviewer).

More specific comments follow:

Results

(lines 151-3) The authors state that "On average 61 (\pm 14%, SD) of the predicted genes in each sample were exclusively found in the present dataset, which highlights the unique gene content of the bathypelagic microbiome". Yet it has long been known that this was to be expected, since deeper sampling results in more genes detected. The authors own recent work recapitulates this well-known fact clearly. (Env. Micro. 2020, Sequencing Effort Dictates Gene Discovery in Marine Microbial Metagenomes. doi: 10.1111/1462-2920.15182.). The fact that more genes were detected with this deeper sampling was expected. The most interesting question here is:

“What is the nature of the novel deep-sea genes, and how do they differ from genes found elsewhere?” Unfortunately, this manuscript does not address, nor answer, this fundamental question in a quantitative nor convincing way.

RESPONSE 11: The sequencing effort done in our deep ocean bathypelagic samples (3.5 Gbp/sample) was 8 times lower than what was applied in the surface oceans by the *Tara* Oceans consortium (30 Gbp/samples), so the reviewer consideration is incorrect. And despite that, we detected a significant proportion of genes present only in our deep ocean dataset when we compared both datasets. Our paper does describe the distribution of this novelty in figure 1C (the Malaspina-only genes labelled in blue and Sample-specific genes in grey). In our previous version we did not display many details on the functional annotation of these novel genes because most of them, specifically 63%, are “unknowns”, and therefore they do not have a functional prediction. Now, based on this reviewer’s comment **we did an extra analysis presenting some extra detail on the functional annotation of this 37% of the genes (Table S2, Figure S2 in the current ms)** that has a functional annotation and, with this analysis, we have included a new paragraph in the current manuscript version (Lines 150-156). Among the KEGG categories, the most abundant genes within this proportion of novel genes were those related to transporters (a 6%) in which two-component systems and ABC transporter were highly abundant. The second was the novel genes related to DNA repair and recombination with 2.5% followed by enzymes within the peptidases which represented over 1.8% of the novel genes. Other genes with representation > 1% were secretion systems, aminoacidic related enzymes or quorum sensing genes (Fig. 1, Fig S2 in the ms).

Figure 1. Histograms showing the genes with functional annotation based on KEGG metabolism hierarchy III (KO) of the novel genes of the Malaspina Gene DataBase (M-GeneDB). This represents 37% of the novel genes of this catalogue since the rest has no functional annotation and are only shown those with abundances > 5000 counts.

Taxonomic affiliations (lines 162-186)

This section is a bit of a recap of the authors previous work and results on the very same samples, and so are not really new, nor surprising or novel, see in: Molec Ecol. 2020. “Major imprint of surface plankton on deep ocean prokaryotic structure and activity”, doi: 10.1111/mec.15454.; PNAS 2018 “Sinking particles promote vertical connectivity in the ocean microbiome” doi: 10.1073/pnas.1802470115.; ISME J. 2016, “Global diversity and biogeography of deep-sea pelagic prokaryotes”. doi: 10.1038/ismej.2015.137.

This previously published work by these same authors makes this section at best confirmatory.

Likewise, information on partitioning of taxa into “free-living” FL (0.2-0.8 um size fraction) or “particle-attached” (PA, 0.8-20um size fraction) fractions is a recapitulation of previous work published by these same authors. (ISME J. 2016, Large variability of bathypelagic microbial eukaryotic communities across the world's oceans, doi: 10.1038/ismej.2015.170.)

RESPONSE 12: We agree with the reviewer that this section confirmed our previous findings except for the case of the analyses of the giruses that was not presented in any previously published study of the bathypelagic deep ocean. Nevertheless, we present these taxonomic assignments for prokaryotes and picoeukaryotes because they are based on metagenomic 16S/18S reads respectively, extracted from our metagenomes, instead of from PCR amplicon tags as in our previous published papers. We still believe that the scientific community would appreciate this summary section showing the relative abundance and distribution of the main taxonomic groups of eukaryotes, prokaryotes, giruses and viruses in the text, although we have moved most of this section as supplemental results and discussion. That the results are similar to those obtained by tag-sequencing is reassuring, and worth mentioning.

Functional architecture of the deep ocean microbiome (lines 187-278)

The beginning of this section focuses on the difference between large and small size fractions. Not surprisingly, the biggest differences were found not biogeographically, but between different size fractions for taxa, and genes. This is an expected result, seen in other contexts and depths as well. This expected result does not really distinguish the deep ocean from other habitats in the water column, and in fact the authors have already published on it, with the very same samples. In their Molec. Ecol. 2015 paper, “Particle-association lifestyle is a phylogenetically conserved trait in bathypelagic prokaryotes” (doi: 10.1111/mec.13419), using the very same samples as in this Nat. Comms submission (from 2150-4000 m) they stated: “As a consequence, PA and FL communities had clear alpha- and beta-diversity differences that exceeded the global-scale geographical variation”. So, **the great differences between large vs. small size**

fraction results reported here, are nothing new. And as taxa vary, so do their genes, as expected.

RESPONSE 13: As the reviewer indicates, our findings could be expected given our previous results with the taxonomic data and other published studies. However, the assumption that as the taxa vary, so do their genes, is not evident nor straightforward (see the studies by Louca et al. 2016a, 2016b that indicate that taxonomy and functional structure might be decoupled in microbial communities). It is well known that genomes with identical 16S rRNA sequences may contain highly different functional gene content and the other way around, in which divergent phylogenetic taxa may display a similar functional gene pool (convergent evolution). Thus, our work constitutes again the first large scale assessment of differences in the metabolic capability of free-living and particle attached microbial assemblages in the deep ocean.

- Louca, S., S.M.S. Jacques, A.P.F. Pires, J.S. Leal, D.S. Srivastava, L.W. Parfrey, V.F. Farjalla, and M. Doebeli. 2016. *High taxonomic variability despite stable functional structure across microbial communities*. *Nature Ecology & Evolution*. **1**: 0015. doi: 10.1038/s41559-016-0015
- Louca, S., L.W. Parfrey, and M. Doebeli. 2016. *Decoupling function and taxonomy in the global ocean microbiome*. *Science*. **353**: 1272-1277.

Lines 208-212. The authors state “To explore the potential metabolic differences between FL and PA prokaryotic microbial communities (Fig.3 and Fig.S4), a selection of marker genes (Table S6) were searched in the bathypelagic metagenomic dataset and their abundances normalized using recA”. They find that among these 49 KEGG orthologs are present.

The authors do not tell us however why these marker genes were selected? Are they the most abundant genes? Are they more abundant than genes expected for heterotrophy? Are they most representative of the community? Are they otherwise somehow statistically significant? Lacking further explanation, the reader is left to assume genes were selectively picked, to support the authors presumptions. Inspection of Table S6 shows a somewhat biased and very specific selection of genes, many of them known to commonly occur in deep-sea bacteria or archaea (including most of those for chemolithotrophic energy generation, anaerobic respiration, or CO₂ fixation in Table S6.)

RESPONSE 14: As we explained in the manuscript, the selection of marker genes that we used included a total of >80 biochemically-relevant genes described previously in the deep ocean related to carbon, nitrogen, sulfur and methane metabolisms. For this work, we focused mostly on these metabolisms rather than other functional traits given our interest in exploring the prevalence of chemolithoautotrophic and mixotrophic metabolisms in the bathypelagic ocean. We decided to use only key genes representative of these pathways that in most cases were associated with one functional module, since many other genes are involved in different pathways, and using all these genes would bias significantly our results. Although these pathways have been described before as stressed by the reviewer, it is the first time that they are assessed at a large-scale bathypelagic ocean sampling at the genome level as we did use our Malaspina Deep MAGs catalogue. All previous studies were local studies without reconstruction of MAGs, so the novelty to link certain metabolic potential to the taxonomy at the genome level is a unique feature of this study. Chemoautotrophy has been described before in the

deep ocean, of course! But our paper identifies genomes from diverse lineages from the bathypelagic ocean that have this potential and explores their biogeography throughout the global tropical and subtropical deep ocean.

Lines 215-278. This section describes the (narrow) set of marker genes the authors chose to focus on, mostly in terms of their % representation in the large or small size fractions, or the different stations sampled. If chemoautotrophy is so prevalent, then how prevalent among the metagenome are these marker genes? We are not really told how abundant these marker genes are, relative to other relevant metabolic pathways, for example those involved in heterotrophy. Even if chemoautotrophic pathway genes are there, how frequently are they (or they products) being expressed and used? This in itself renders the argument for the general importance for chemoautotrophy in the deep ocean, somewhat moot. The idea of “mixotrophy” is discussed, but this concept, more accurately termed facultative chemoautotrophy, is well known, and should probably be better reference. Some genes, like Rubisco, are used for other purposes than CO₂ fixation as well, which the authors do admit.

RESPONSE 15: We have explained above the prevalence of chemoautotrophy based on the presence of individual genomes (MAGs) that contain these marker genes (RuBisCo and other genes). This approach is much more reliable than gene centric approaches because you can link much better function with taxonomy, explore the pathway completeness, and the pool of genes involved in energy supply. Indeed, **we mentioned that the potential mixotrophy (or facultative chemoautotrophy as termed by the reviewer) is present in about 22% of the samples based only on trustful MAGs that contain the form I or II RuBisCo genes.** Those RuBisCo forms are the ones linked to autotrophy and, for that reason we built a phylogeny (**Figure S7**) and accounted only for those MAG genomes containing the RuBisCo form I and II to estimate the potential chemoautotrophy based on the Calvin Benson-Bassham cycle. We have revised the text in our manuscript to clarify our analysis (lines 443-444; 463-465. Although we agree with the reviewer that knowing the relative importance of chemoautotrophy versus heterotrophy would be very interesting, this comparison is not trivial since many genes involved in heterotrophy may participate in different pathways, and thus their relative abundance cannot be properly accounted for.

Diversity and novelty of the Malaspina Deep MAGs catalogue. (lines 280-366)

These final sections discuss the analyses of 317 medium quality or better MAGs, assembled from the dataset. **These data are certainly interesting, but they do not seem to provide much integrative or compelling conclusions broadly, with respect to the deep ocean microbiome.**

Genome resolved metabolic capabilities of the bathypelagic ocean (lines 338-504)
A handful of the MAGs (25) are shown to contain some of the marker genes for nitrification, denitrification, nitrogen fixation, sulfur oxidation and so on – and this is discussed in some detail. But most of these genes and pathways are already known for deep-sea marine bacteria, with respect to the pathways, and many of the taxa that contain them. Again, these are interesting data, but not entirely compelling for making grand conclusions regarding the metabolic architecture of the deep-ocean microbiome.

In conclusion, this is a certainly a great dataset, and there are some really interesting findings, but the authors have taken a somewhat narrow approach in their analysis, with a somewhat biased focus on just a few hand selected pathways – which is not a broad description of “metabolic architecture”. A large focus of the paper, the differentiation of large (“PA”) and small (“FL”) size fraction taxa and genes. Are there diverse taxa and metabolic strategies on particles, including chemoautotrophy? Yes, this is already well known, supported by this work. But it is not specific to the deep-sea, and has already been noted by these authors and others. Finally, the potential for chemolithotrophy (and anaerobic respiration, and nitrogen fixation) in deep-sea bacteria and archaea has been known for some time, and described in previously published works. **One of the authors central claims in the abstract, that “Interestingly, the potential to grow both autotrophically and heterotrophically was a prevalent prokaryotic strategy...” simply is not well supported by the data and analyses. The most abundant carbon fixation marker gene, ribulose-bisphosphate carboxylase (which may be involved in processes other than autotrophy as well) was found in only ~1.3% of the cells. This does not make a very compelling case (along with simple thermodynamic considerations), for the centrality of chemolithotrophy that the authors put forward.**

RESPONSE 16: The reviewer considers that our claim that chemoautotrophy is relevant in the deep ocean is not correct. He uses as evidence the 1.3% average number of cells cited above, and other similar comments that we already responded to above (see Response 10). Besides, we have done a rank abundance plot to better visualize that some of the MAGs containing genes for chemolithoautotrophy are abundant members of the deep ocean. We performed a rank abundance curve (**Figure 2**) with the accumulated abundance of 16 MAGs with clear genetic potential for chemolithoautotrophy that includes the Ammonia Oxidizing Archaea (AOA), Ammonia Oxidizing Bacteria (AOB), Nitrite Oxidizing Bacteria (NOB), Sulfur Oxidizing bacteria (SOB) and MAGs containing the pathway for inorganic carbon fixation of the 3-Hydroxypropionate (3-HP) or MAGs with the RuBisCo Forms I and II genes. **Figure 2** below, that has been included as **Figure S8 in the current version of the ms**, shows clearly that many of these potential chemolithoautotrophic MAGs are within the top 50 most abundant genomes of the 317 MAG within our deep bathypelagic dataset. MAG0176 (NOB) is in the top 25, MAG0052 containing the RuBisCo Form I gene with a 35% Calvin Benson-Bassham cycle (CCB) pathway completeness is in the top 29, MAG0166 with the RuBisCo Form II gene and 43% (CCB) pathway completeness is in the top 42 and so on (see the rest on **Table S14**). These extra results show **that many of these MAGs with potential chemolithoautotrophy belong to the abundant taxa of the bathypelagic deep ocean from the tropical and temperate oceans.** All this extra information has been included now in the ms. in lines 466-469.

Figure 2. Rank abundance curve presenting the accumulated abundance of the 317 MAGs based on reads per kilobase per genome equivalent (RPKGs) and colouring the 16 MAGs with identified genetic potential for chemolithoautotrophy. The MAGs with chemolithoautotrophy potential are the Ammonia Oxidizing Archaea (AOA), Ammonia Oxidizing Bacteria (AOB), Nitrite Oxidizing Bacteria (NOB), Sulfur Oxidizing bacteria (SOB) and MAGs containing pathways for inorganic carbon fixation such as the 3-Hydroxypropionate (3-HP) or MAGs with the RuBisCo Forms I and II genes associated to autotrophy.